# Systematic HOIP interactome profiling reveals critical roles of linear ubiquitination in tissue homeostasis

Yesheng Fu[1,4], Lei Li[1,4], Xin Zhang [1,4], Zhikang Deng[1,4], Ying Wu[1,4], Wenzhe Chen[1], Yuchen Liu[1], Shan He[1], Jian Wang [1], Yuping Xie[1], Zhiwei Tu[1], Yadi Lyu[1], Yange Wei[1], Shujie Wang[1], Chun-Ping Cui [1], Cui Hua Liu [2,3] ✉ & Lingqiang Zhang [1] ✉

Linear ubiquitination catalyzed by HOIL-1-interacting protein (HOIP), the key component of the linear ubiquitination assembly complex, plays fundamental roles in tissue homeostasis by executing domain-specific regulatory functions. However, a proteome-wide analysis of the domain-specific interactome of HOIP across tissues is lacking. Here, we present a comprehensive mass spectrometry-based interactome profiling of four HOIP domains in nine mouse tissues. The interaction dataset provides a high-quality HOIP inter-actome resource with an average of approximately 90 interactors for each bait per tissue. HOIP tissue interactome presents a systematic understanding of linear ubiquitination functions in each tissue and also shows associations of tissue functions to genetic diseases. HOIP domain interactome characterizes a set of previously undefined linear ubiquitinated substrates and elucidates the cross-talk among HOIP domains in physiological and pathological processes. Moreover, we show that linear ubiquitination of Integrin-linked protein kinase (ILK) decreases focal adhesion formation and promotes the detachment of *Shigella flexneri*-infected cells. Meanwhile, *Hoip* deficiency decreases the linear ubiquitination of Smad ubiquitination regulatory factor 1 (SMURF1) and enhances its E3 activity, finally causing a reduced bone mass phenotype in mice. Overall, our work expands the knowledge of HOIP-interacting proteins and provides a platform for further discovery of linear ubiquitination functions in tissue homeostasis.

Linear ubiquitination (also known as Met1-linked ubiquitination) has emerged as critically important roles for maintaining homeostasis by controlling vital signaling cascades[1]. Dysregulation of this process is associated with imbalance of homeostasis and severe pathologies, including immune disorders, angiogenesis defects, cancer and neurodegeneration[2,3]. Assembly of Met1-linked ubiquitin chains is mediated by Linear Ubiquitination Assembly Complex (LUBAC), the only known ubiquitin ligase in vertebrates. LUBAC is composed of HOIL-1-interacting protein (HOIP), Haem-oxidized IRP2 ubiquitin ligase 1 (HOIL-1) and Shank-associated RH domain interacting protein

[1]State Key Laboratory of Medical Proteomics, National Center for Protein Sciences (Beijing), Beijing Institute of Lifeomics, Beijing 100850, China. [2]CAS Key Laboratory of Pathogenic Microbiology and Immunology, Institute of Microbiology, Chinese Academy of Sciences, Beijing 100101, China. [3]Savaid Medical School, University of Chinese Academy of Sciences, Beijing 101408, China. [4]These authors contributed equally: Yesheng Fu, Lei Li, Xin Zhang, Zhikang Deng, Ying Wu. ✉e-mail: liucuihua@im.ac.cn; zhanglq@nic.bmi.ac.cn

(SHARPIN)[4–8]. Numerous studies involving biochemical and genetic methods have shown that LUBAC-mediated linear ubiquitination controls immune signaling, NF-κB transcription factor activation, receptor activation and cell death[6,9–11]. However, an in-depth understanding of how LUBAC integrates with cellular protein-protein interaction (PPI) networks is lacking, and such information would be valuable for providing insights into the systematic physiological roles of linear ubiquitin chains.

HOIP, the only catalytic component of LUBAC, has remarkable specificity and exclusively assembles linear ubiquitin chains. The identification of key HOIP-interacting partners with genetic mouse models has greatly facilitated our understanding of crucial functions of linear ubiquitination in tissue homeostasis. Complete HOIP deficiency causes embryonic lethality with vascular defects[9,11]. HOIP deficiency in liver parenchymal cells triggers tumorigenesis with spontaneous hepatocyte apoptosis and liver inflammation[12]. Treg cell-specific ablation of HOIP leads to severe Treg cell deficiency and lethal immune pathology while mice with B-cell-specific deletion of HOIP show impaired B1-cell development[13,14]. Deletion of epithelial HOIP during influenza A virus (IAV) infection worsens lung injury and decreases survival[15]. Defective HOIP catalytic activity in macrophages, but not in intestinal epithelial cells, ameliorates dextran sodium sulfate-induced colitis[16]. Moreover, bacteria and viruses always hijack linear ubiquitination within host cells to cause the disturbance of tissue homeostasis[1]. For example, *Shigella flexneri* targets HOIP for degradation to invade host cells[17]. However, most studies regarding the functional elucidation of HOIP in tissue homeostasis have been based primarily on certain cell types or organs of interest, thereby limiting the generalization and comparison of the findings to other tissues, such as stomach and bones. Furthermore, none of the previous studies have used proteomics methods, such as affinity purification coupled to mass spectrometry (AP-MS)[18], to systematically examine physiological functions of linear ubiquitination in tissue homeostasis.

HOIP contains several domains: the peptide: N-glycanase /ubiquitin-associated UBA-containing or UBX-containing (PUB) domain, the zinc finger-Npl4 zinc finger (ZF-NZF, also referred to as NZF) domain, the ubiquitin-associated (UBA) domain, the Really Interesting New Gene (RING)-in-between-RING (RBR) domain and the linear ubiquitin chain determining (LDD) domain[1]. Deubiquitinases, including OTU deubiquitinase with linear linkage specificity (OTULIN) and cylindromatosis (CYLD) directly or indirectly interact with the PUB domain of HOIP to remove the linear ubiquitin chains catalyzed by LUBAC[19]. NZF is considered as a domain for substrate recognition[20]. For example, HOIP conjugates linear ubiquitin chains onto NEMO and ALK1 through the NZF domain[2,7]. UBA is essential for the recruitment of HOIL-1 and SHARPIN to modulate the activity of LUBAC[21]. RBR-LDD determines the activity and specificity of LUBAC toward substrates[22–24]. A few reports have indicated that mutation or truncation of these domains can affect certain functions of LUBAC[4,5,25,26], but no systematic comparative assessment of the contributions of individual HOIP domains to LUBAC functions has been conducted.

In this work, we performed systematic AP-MS analysis through GST pull-down assays using four truncations of HOIP (including PUB, NZF, UBA and RBR-LDD domains) across nine mouse tissues (including brain, heart, liver, spleen, lungs, kidneys, stomach, colorectum and bones), and we identified an average of approximately 90 interactors for each bait in a tissue. Using this knowledgebase, we found that tissue-shared HOIP PPIs were associated with important cellular events and PPIs in a single tissue were linked to tissue-specific functions and diseases. Notably, the domain interactome identified a set of previously undefined substrates of linear ubiquitination and elucidated the cross-talk among HOIP domains in physiological and pathological processes. Further analysis of HOIP PPIs related to kinases and ubiquitination showed that linear ubiquitination of Integrin-linked kinase (ILK) promoted cells defense *Shigella flexneri* infections and that

linearly ubiquitinated SMAD-specific E3 ubiquitin-protein ligase 1 (SMURF1) promoted bone formation by disrupting the interaction between ubiquitin-conjugating enzyme E2 and SMURF1. Overall, our results provide systematic insights into the interaction patterns of HOIP, highlighting the crucial roles of linear ubiquitin chains in physiological and pathological processes.

## Results

### Systematic mapping of HOIP protein-protein interactions in mouse tissues

To identify the HOIP interactome across tissues, including the brain, heart, liver, spleen, lungs, kidneys, stomach, colorectum and bones, GST only, GST-HOIP PUB, NZF, UBA and RBR-LDD truncated proteins were expressed in BL21 bacteria and purified for next mouse-tissue affinity purification and mass spectrometry (Fig. 1a). Consistent with previous reports, we found that OTULIN specifically bound to the PUB domain of HOIP and SHARPIN specifically interacted with the HOIP UBA domain in all tissues (Supplementary Fig. 1a). The mass spectrometry data of HOIP interacting proteins were then generated from three replicates of each truncation in one tissue. Firstly, the raw data were filtered stringently with a 1% false discovery rate (FDR) (Supplementary Fig. 1b and Supplementary Data 1). Next, pull-downs of each HOIP-truncated protein versus GST alone were calculated and the fold-over-control (FOC) data were integrated with p value analysis by using iBAQ under normalization and imputation. With cutoffs of an FOC ≥ 2 and a $P_{FOC} \leq 0.05$, we identified more than 80% of reported HOIP interacting proteins, whereas an FOC cutoff greater than two only slightly increased the number of reported HOIP-interacting proteins detected in the screening (Supplementary Fig. 1c). Therefore, we defined statistically significant HOIP PPIs by setting the cutoff values such that FOC ≥ 2 and $P_{FOC} \leq 0.05$. This filtering strategy removed many common false-positive interacting proteins (Supplementary Fig. 1d). In total, we uncovered 17078 proximity interactions representing 3430 non-redundant proteins, an average of approximately 90 interactors for each bait in a tissue (Fig. 1b and Supplementary Data 2), and the numbers of identified interactions varied significantly across domains and tissues (Fig. 1c, d). Next, we recovered approximately 150 HOIP-interacting proteins in the Biological General Repository for Interaction Datasets (BioGRID), which was more than other large-scale AP-MS studies, including BioPlex 2.0 and BioPlex 3.0, as well as in the literature-curated interaction dataset (Fig. 1e); this supported the reliability of our data. Extensively studied proteins (as determined by the number of citations in PubMed) had, on average, significantly more annotated HOIP-interacting proteins in the literature-curated interaction dataset. However, our data showed no such differences (Supplementary Fig. 1e), supporting the unbiased character of our data.

Linear ubiquitination regulates multiple signaling pathways, cell death and proteasome function, which are vital for organism homeostasis[1,27]. Here, we found that our data were consistent with previous studies. Specifically, the majority of the components previously classified as being associated with NF-κB, PI3K, JAK-STAT signaling, and the proteasome complex were present in our results (Fig. 1f). Interestingly, several reported HOIP interactors (e.g., STAT1 and β-Catenin)[28–30] showed tissue-specific interactions in our data (Fig. 1g), which was confirmed by GST pull-down and endogenous co-immunoprecipitation assays in different tissues (Fig. 1h, i and Supplementary Fig. 1f). STAT1 showed strong interaction with HOIP in the liver, lungs and colorectum, suggesting that linear ubiquitination of STAT1 is crucial for immune homeostasis in these tissues (Fig. 1h). The interactions between β-Catenin and HOIP were obviously detected in the brain, liver, lungs, stomach and colorectum, and these interactions were probably due to the specific expression of β-Catenin in these tissues or the specific binding of

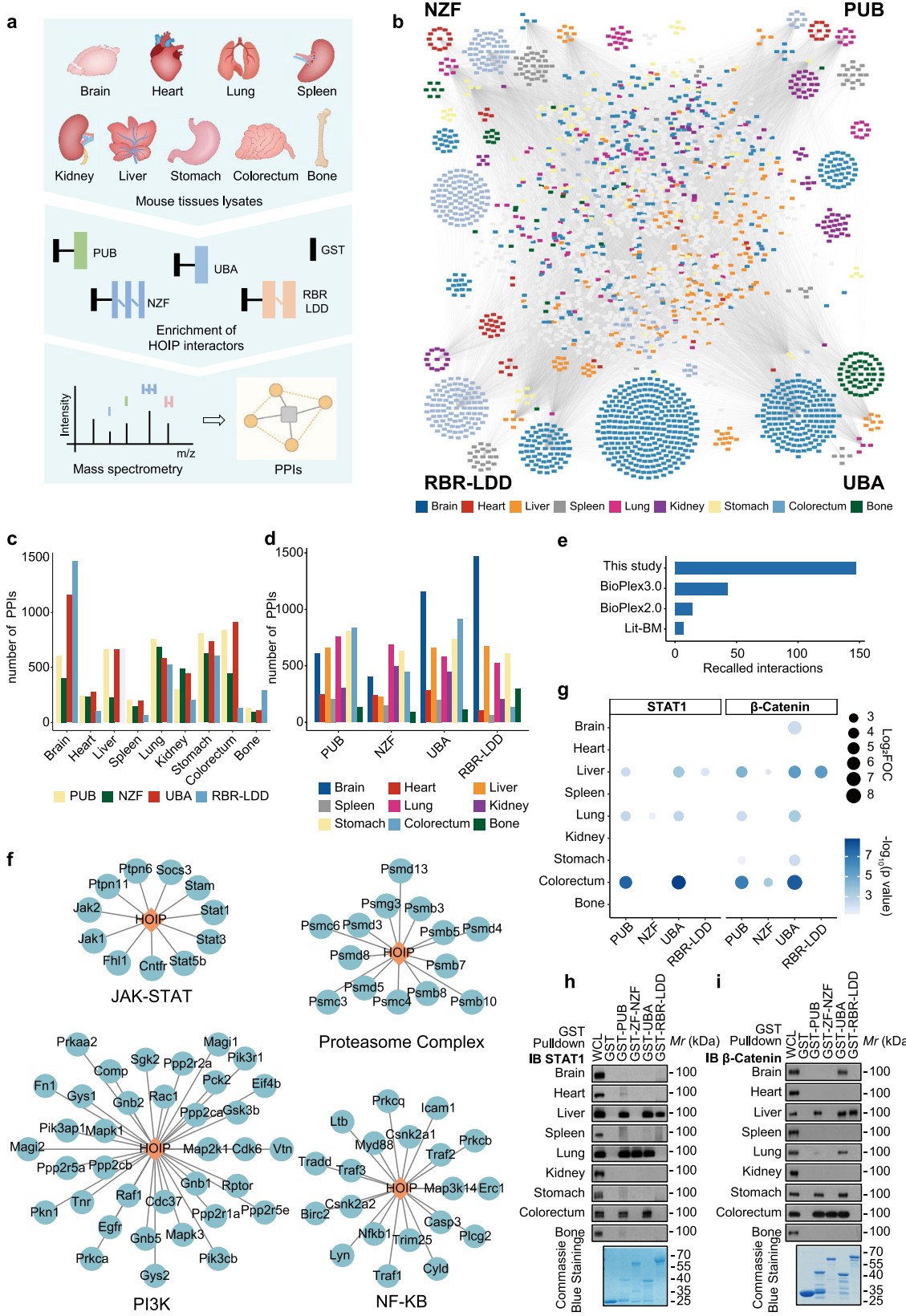

HOIP to FXR1 (FMR1 autosomal homolog 1), a known β-Catenin's interactor[31] (Fig. 1i and Supplementary Fig. 1g, h). Together, these results suggest that the HOIP-interacting protein network is well-defined and show that patterns of HOIP domain and tissue specific interactions can be effectively used to speculate functional associations.

## Identification of tissue-shared and tissue-specific HOIP PPIs

Next, we further analyzed the patterns of known HOIP-interacting proteins across tissues in our data. Gene Ontology (GO) enrichment analysis of these interactions associated with known functions of HOIP showed different tissue distributions. For example, interactions associated with the regulation of immune effector process were specifically

**Fig. 1 | Quantitative HOIP interactome profiles in mouse tissues. a** Workflow used to generate the four HOIP domains interaction network in nine tissues. This includes 45 GST pull-downs, triplicate runs on a hybrid linear ion trap-Orbitrap mass spectrometer. **b** Protein-protein interactions (PPIs) network of four HOIP domains across nine tissues. Preys are shown as squares and tissue-specific preys are indicated by colors. **c** Numbers of HOIP PPIs quantified across tissues. **d** Numbers of HOIP PPIs quantified across domains. **e** Numbers of all HOIP PPIs detected in different Affinity purification coupled to mass spectrometry (AP-MS) studies that are also reported by additional studies deposited in the BioGRID database. **f** Representative HOIP PPIs that are recapitulated in the generated network. **g** Dot plot of changes of HOIP interacting proteins STAT1 and β-Catenin across tissues and domains. Size of dot represents protein abundance relative to GST. Color of dot represents $p$ value between domain and GST. $p$ values are from the Limma moderated $t$-test. Immunoprecipitation analysis of known interacting proteins, such as STAT1 (**h**) and β-Catenin (**i**) in domains and tissues. The GST alone protein was a negative control. Source data are provided as a Source Data file.

enriched in the spleen, a canonical immune organ, while NF-κB signaling and apoptosis were widely distributed across tissues (Supplementary Fig. 2a and Supplementary Data 3). Indeed, we found that the interaction of OTULIN or SHARPIN with HOIP was constitutive and observed in all tissues (Supplementary Fig. 2b), suggesting that a portion of the interactions were tissue-shared. Meanwhile, the interactions for tissues (calculated by the Jaccard index) correlated very well (Fig. 2a and Supplementary Data 3), prompting us to identify tissue-shared and tissue-specific HOIP PPIs. We defined PPIs as tissue-shared if these PPIs were detected in at least six tissues. In contrast, if PPIs were detected in only one tissue, those PPIs were considered potentially tissue-specific. In total, we observed 109 (3.1%) tissue-shared PPIs and 1817 (52.9%) potential tissue-specific PPIs (Supplementary Data 4). The brain was found to have the largest proportion of potential tissue-specific PPIs, followed by the liver, bone and colorectum (Fig. 2b). Functional analysis of tissue-shared PPIs showed that these PPIs were mainly involved in basic cell activities such as chromosome organization, endocytosis, cellular respiration and small molecule catabolic processes (Fig. 2c and Supplementary Data 4). RUVBL1 and RUVBL2 are ATP-dependent chromatin remodelers that serve to regulate DNA accessibility by repositioning, ejecting, or modifying nucleosomes[32]. We found that several RUVBL1/2 related complexes were tissue-shared PPIs, and subsequent coimmunoprecipitation (CO-IP) analysis showed that RUVBL1/2 bound to HOIP in all tested tissues (Fig. 2d), indicating an undefined role of linear ubiquitination in chromosome remodeling. OTULIN, the deubiquitinase of linear ubiquitination, antagonizes SNX27-dependent cargo loading and endosome-to-plasma membrane trafficking[33]. However, we showed that adaptor protein complexes, such as AP2B1 and AP2M1 (which connected transmembrane protein cargos to cellular vesicular coats) were HOIP tissue-shared PPIs, as confirmed by CO-IP. This indicates that linear ubiquitination might participate in another process of cargo translocation (Supplementary Fig. 2c).

The enrichment of HOIP PPIs that were specifically enriched in one tissue was presented in Fig. 2e and Supplementary Data 4. As expected, the response to tumor necrosis factor was highly enriched for the spleen-specific PPIs, probably due to the specific interaction between HOIP and TRAF1/2 (Fig. 2e and Supplementary Fig. 2d), consistent with the findings of a previous study[34]. Mice with hepatocyte-specific OTULIN deletion showed an almost complete lack of glycogen in their livers, suggesting a role of linear ubiquitination in metabolism[35]. HOIP liver-specific PPIs participated in ribose phosphate metabolic processes, probably by modulating certain regulators, such as sterol carrier protein 2 (SCP2) and AMP-activated protein kinase catalytic subunit alpha-2 (PRKAA2) (Supplementary Fig. 2d), which further supports a metabolism-regulatory function of linear ubiquitination. Recently, linear ubiquitination was found to promote proteasomal degradation of toxic, disease-associated protein aggregates in neurodegenerative diseases[36]. Our brain-specific PPIs identified some functional proteins associated with synapse organization, such as neuronal proteins synapsin-1 (SYN1) and huntingtin-associated protein 1 (HAP1) (Supplementary Fig. 2d), revealing a previously undefined regulatory pattern of linear ubiquitination that affects the physiological function of the brain. Similarly, analysis of tissue-specific PPIs showed that cadherin-related family member 5 (CDHR5), which is

associated with microvillus organization and ATP-dependent translocase ABCB1, which is involved in the osmotic stress response, were colorectum-specific (Supplementary Fig. 2d), suggesting a function of linear ubiquitination in the colorectum that has not been identified by existing studies. Moreover, we investigated whether HOIP PPIs in tissues were associated with particular genetic diseases by using the Online Mendelian Inheritance in Man (OMIM) database[37]. A cluster of significant OMIM disease terms were found to be tissue-specific ($p < 0.05$, Fisher's exact test, Fig. 2f and Supplementary Data 4). Notably, heart-specific HOIP PPIs annotated with cardiomyopathy were highlighted by the OMIM analysis ($p = 1.01 \times 10^{-5}$) (Supplementary Fig. 2e). Among these disease-related PPIs, the interaction of MYL3 and CSRP3 was confirmed by CO-IP analysis and showed heart specificity (Supplementary Fig. 2d). Although previous studies have revealed that HOIP deficiency in the endothelium results in cardiovascular failure due to aberrant TNFR1-mediated endothelial cell death[11], in cardiopathy, the HOIP interacting proteins MYL3 and CSRP3 have been suggested to be other regulatory targets of HOIP in cardiac muscles. These findings suggest additional disease-associations of linear ubiquitination. Mitochondrial complex I deficiency is characterized by defects in oxidative phosphorylation, but patients' symptoms vary widely in nature and severity, and there is no cure[38]. HOIP PPIs analysis showed that the mitochondrial complex I deficiency was highly enriched in the heart ($p = 4.23 \times 10^{-6}$), bones ($p = 0.0012$) and stomach ($p = 0.0048$), indicating that a coordinated interplay of mitochondrial functions regulated by HOIP in these tissues, providing a potential shared mechanism for mitochondrial complex I deficiency across tissues. In addition, HOIP interacted with proteins associated with other Mendelian diseases, such as cortical dysplasia ($p = 0.0077$) and polycystic kidney disease ($p = 0.0081$), which has not been reported previously.

Overall, tissue-shared PPIs were found to be involved in fundamental cellular events, such as chromosome remodeling and endocytosis, suggesting indispensable roles of linear ubiquitination. Assessment of tissue-specific PPIs revealed brain-specific synapse organization, liver-specific ribose phosphate metabolic processes, heart-specific myocardial function and other features, providing a systematic understanding of linear ubiquitination functions in every tissue. OMIM analysis identified tissue-specific diseases and multi-tissue-related diseases, further signifying the broad functional roles of HOIP-mediated linear ubiquitination across tissues.

**Comparison of HOIP domain-mediated PPI networks**

To identify biological processes and pathways that might be targeted by the four HOIP domains, enrichment analysis was performed on sets of proteins interacting with individual domains. A large group of HOIP PPIs involved in fundamental cellular functions, such as the cell cycle, cell morphology, cell death, metabolism, RNA regulation and translation, were identified in more than two domains (Supplementary Fig. 3a and Supplementary Data 5). This was consistent with the enrichment proportions of HOIP PPIs across domains. Approximately 50% of the detected HOIP PPIs were found across the four domains (Supplementary Fig. 3b), probably due to the indirect interactions of HOIP domains and certain protein complexes. Further analysis confirmed that networks of the four HOIP domains were enriched in proteins

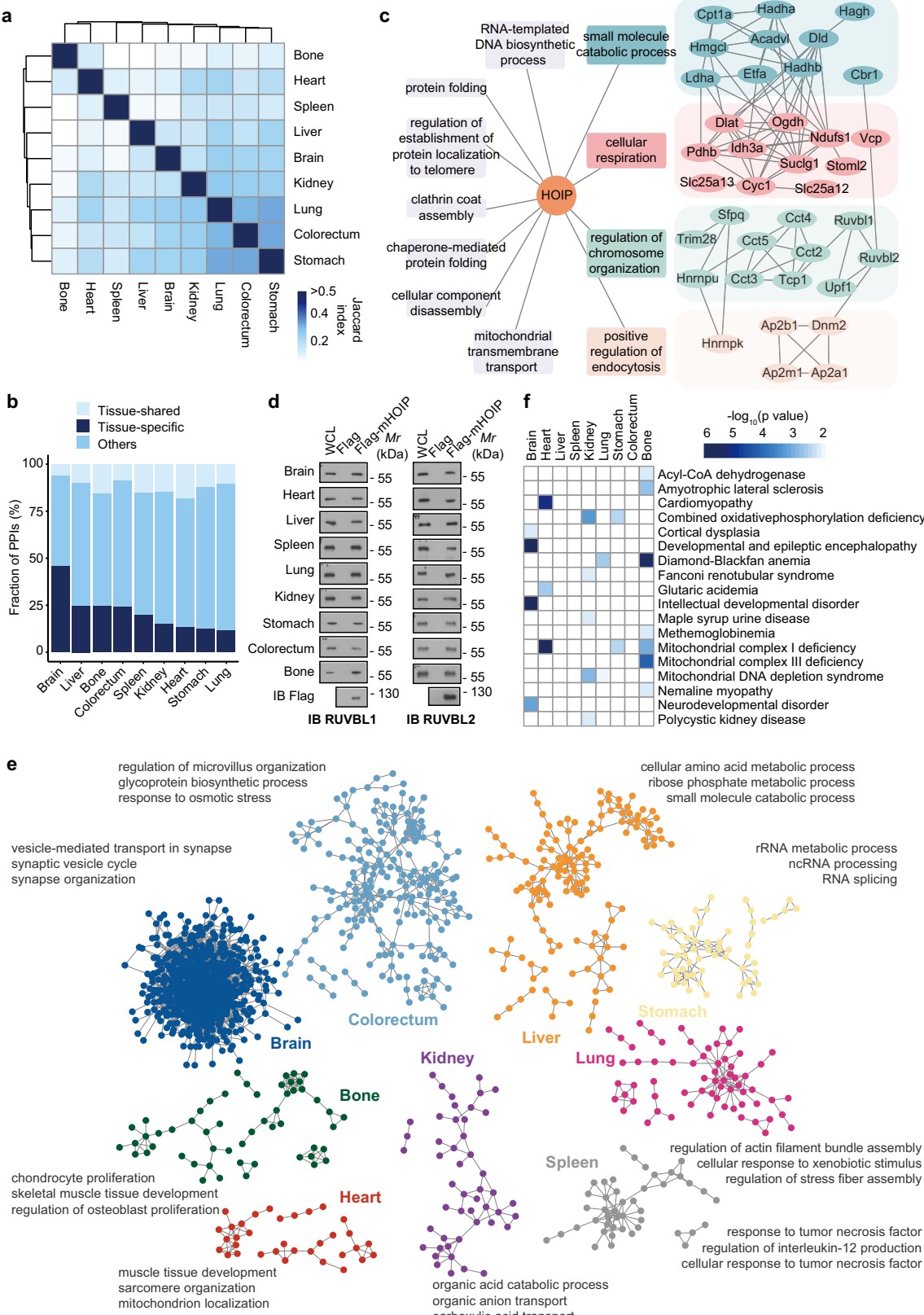

**Fig. 2 | Identification of tissue-shared and tissue-specific HOIP PPIs. a** Jaccard index of HOIP protein-protein interactions (PPIs) detected by various tissues to show PPIs coverage across tissues. **b** Distribution of the fraction of HOIP PPIs quantified across different tissues. **c** HOIP PPIs were analyzed by using gene ontology biological process and cellular compartment categories. Some categories of basic cell activities are shown as interaction network. **d** Immunoprecipitation of ectopically expressed HOIP from HEK293T cells in different tissues and immunoblot with the antibodies of tissue-shared proteins. The ectopically expressed Flag-tagged only peptide was a negative control. **e** Protein enrichments across tissues and their biological functions. The enriched proteins represent tissue-specific HOIP PPIs. **f** Heatmap of the enrichment of genetic diseases across tissues. The disease terms are from the OMIM database. p values are from the Fisher exact test. Source data are provided as a Source Data file.

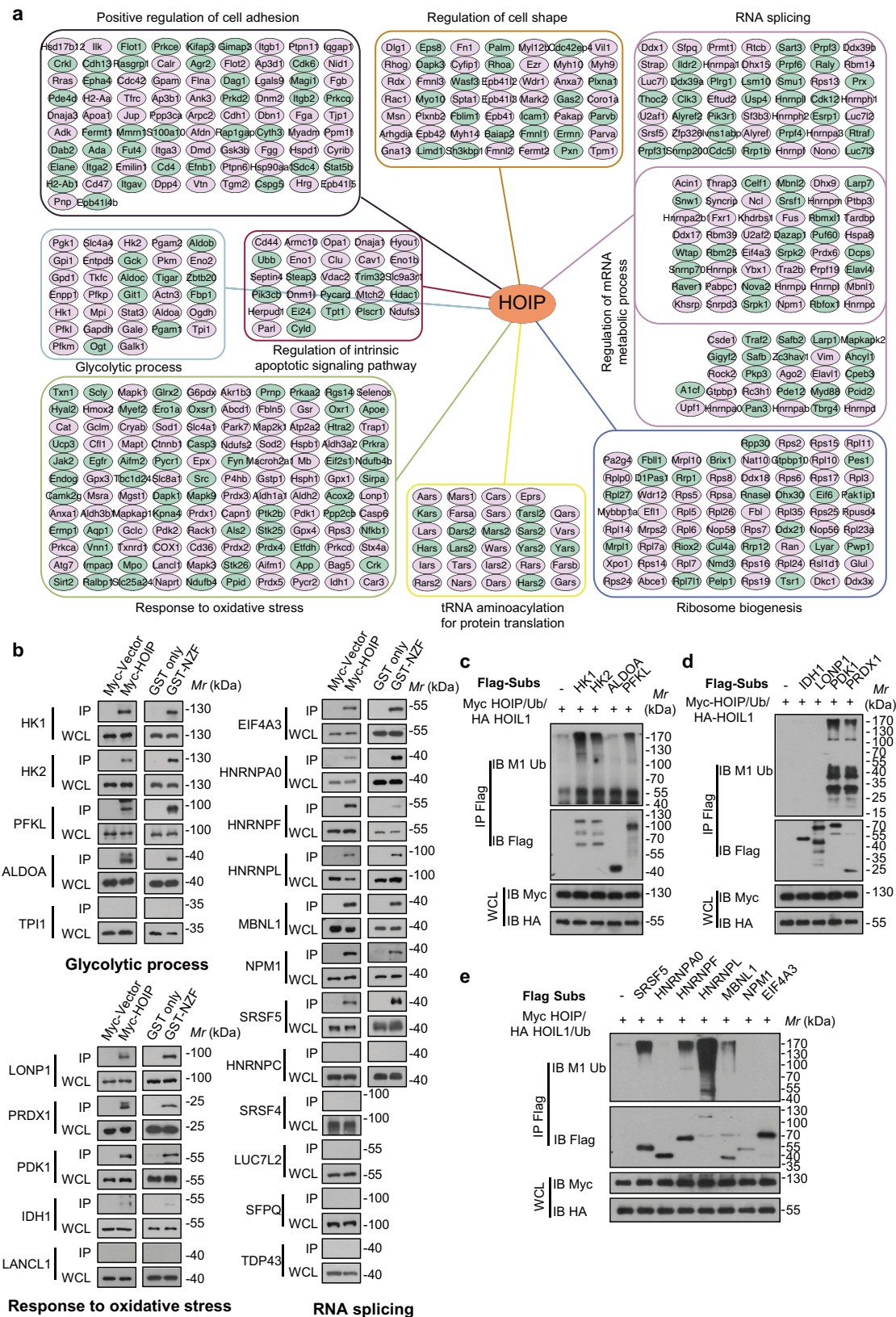

involved in similar biological processes (terms significantly enriched in all domains) (Fig. 3a, Supplementary Fig. 3c and Supplementary Data 5), among which the most represented processes were glycolytic processes, RNA splicing and responses to oxidative stress. In a case study to demonstrate the HOIP PPIs involved in these processes, Myc-tagged HOIP was used to examine the interaction of HOIP with

exogenously expressed proteins that were identified in at least two tissues. Fifteen out of 22 (~70%) HOIP neo interactors were detected in the Myc-tagged HOIP immunocomplexes in HEK293T cells. Moreover, we found that these interactors were HOIP NZF-binding and we confirmed their interactions with GST-HOIP NZF pull-down and Myc-HOIP NZF co-immunoprecipitation assays (Fig. 3b and Supplementary

**Fig. 3 | Comparison of HOIP domain-mediated PPIs networks. a** Summary diagram of similar biological functions (manually selected) of global HOIP interactors. The round rectangles indicate HOIP protein-protein interactions (PPIs) in similar biological functions, the small red nodes indicate HOIP PPIs in one tissue and the small green nodes indicate HOIP PPIs in at least two tissues. **b** Immunoprecipitation of ectopic expressed HOIP and bacteria purified HOIP NZF in lysates from HEK293T cells transfected with indicate tagged proteins, and immunoblot with the corresponding antibodies. The Myc-tagged vector and GST alone protein were used as negative controls. **c** Immunoprecipitates of HK1, HK2, ALDOA and PFKL to detect the linear ubiquitination in HEK293T cells transfected with Myc-tagged HOIP and HA-tagged HOIL-1. **d** Immunoprecipitates of IDH1, LONP1, PDK1 and PRDX1 to detect the linear ubiquitination in HEK293T cells transfected with Myc-tagged HOIP and HA-tagged HOIL-1. **e** Immunoprecipitates of SRSF5, HNRNPA0, HNRNPF, HNRNPL, MBNL1, NPM1 and EIF4A3 to detect the linear ubiquitination in HEK293T cells transfected with Myc-tagged HOIP and HA-tagged HOIL-1. Source data are provided as a Source Data file.

Fig. 3d). The NZF domain of HOIP is previously implicated in regulation of substrates bound for subsequent linear-ubiquitination catalysis via the RBR-LDD domain[2,6,7]. Therefore, linear ubiquitination assays were utilized to confirm whether these candidates were potential substrates. Nine out of 15 (60%) neo-interactors were shown to be ubiquitinated by LUBAC while a HOIP-CS mutant (C699, 702, 871, 874 S) that could not generate linear polyubiquitin failed to ubiquitinate these potential substrates (Fig. 3c–e and Supplementary Fig. 3e, f). HOIP and HOIL-1 mutations have been previously shown to cause amylopectinosis, a form of glycogen storage disease[25,26]. Although a more recent study has reported that HOIL-1 targets unbranched glucosaccharides and is required to prevent polyglucosan accumulation[39], the role of linear ubiquitination mediated by HOIP in metabolic control remains relatively unclear. Here, we found that Hexokinase-1 (HK1), Hexokinase-2 (HK2) and liver-type phosphofructokinase (PFKL) in glycolytic signaling were linearly ubiquitinated, suggesting a potential mechanism of linear ubiquitination in regulating glucose metabolism. Moreover, linear ubiquitination of 3-phosphoinositide-dependent protein kinase 1 (PDK1) and peroxiredoxin-1 (PRDX1) in the oxidative stress response and of serine/arginine-rich splicing factor 5 (SRSF5), heterogeneous nuclear ribonucleoprotein F (HNRNPF), heterogeneous nuclear ribonucleoprotein L (HNRNPL) and muscleblind-like protein 1 (MBNL1) in RNA splicing were all detected. This has not been reported previously and expands the potential cellular functions of linear ubiquitination.

Although PPIs of HOIP domains varied across domains, we mainly focus on HOIP-interacting proteins in only one domain, which we defined as domain-specific PPIs, to further investigate the regulation of HOIP itself (Supplementary Fig. 4a, b and Supplementary Data 5). The PUB domain of HOIP is critical for the deubiquitinase OTULIN/CYLD interaction and OTULIN limits cell death by deubiquitinating LUBAC[40,41]. This might explain why apoptosis was enriched in the PUB domain. LUBAC was previously shown to regulate chromosome alignment in mitosis by targeting CENP-E to attach kinetochores[42]. Here, we found specific enrichment of mitotic spindle organization in the PUB domain, further indicating that OTULIN coordinates with HOIP to function in G2/M phase of mitosis. HOIP is required for autophagy initiation and maturation mediated by linear ubiquitination of ATG13[43,44]. Consistently, we found that these autophagy-related processes were specifically enriched in the RBR domain, the core catalytic domain responsible for linear chain formation. Interestingly, although the UBA domain is essential for the recruitment of HOIL-1 and SHARPIN to modulate the activity of LUBAC[21], the enrichment of protein location in the UBA domain suggests an as-yet-undefined function of this domain. Notably, proteins associated with RNA splicing and metabolic processes were enriched in the RBR domain, as well as the PUB domain or NZF domain, suggesting that these processes were coregulated by HOIP domains. In addition, HOIP domain-specific PPIs were also analyzed in lungs from LPS-induced sepsis models (Supplementary Fig. 5a–d and Supplementary Data 6) and the results showed that LPS treatment upregulated some significant GO terms (such as cellular response to interleukin-7 and defense response to bacterium) and downregulated some GO terms, such as establishment or maintenance of epithelial cell apical/basal polarity.

Altogether, we characterized potential linearly ubiquitinated substrates recognized by the NZF domain that are involved in glycolytic processes, RNA splicing and oxidative stress and identified specific functions of the HOIP PUB, UBA and RBR domains in autophagy, protein localization, RNA splicing and metabolic processes. Moreover, we revealed domain-specific HOIP PPIs in LPS-induced sepsis models.

## LUBAC provides defense against bacterial infection by linear ubiquitination of ILK

Importantly, many of the identified HOIP interactors were crucial regulators of cellular functions, such as E3 ligases, DUB enzymes, kinases and transcription factors (Supplementary Fig. 6a and Supplementary Data 7). Previous studies have suggested that identification of the interactors of E3 ligases by AP-MS was challenged due to the transient interactions and potential ubiquitination mediated proteasomal degradation[45]. Therefore, 27 DUB enzymes, 13 HECT E3 ligases and 47 RING E3 ligases in tissues were found in HOIP-interactome by analysis of original data (Supplementary Fig. 6a and Supplementary Data 7). Ten out of 11 selected ubiquitin associated HOIP-interacting proteins (NEDD4, WWP1, ITCH, SMURF1, SMURF2, TRIM32, TRIM46, RNF213, OTUB1 and USP10) were validated with co-IP experiments (Supplementary Fig. 6b). In addition, the top ten of HOIP-interacting proteins associated with kinases were analyzed (Supplementary Fig. 6c and Supplementary Data 7). Among these kinases, Integrin-linked protein kinase (ILK) in the colorectum was the most significantly enriched HOIP-interacting protein.

Further exogenous and endogenous immunoprecipitation experiments showed that ILK interacted with HOIP (Supplementary Fig. 7a–c). Domain mapping assays confirmed that HOIP interacted with the kinase catalytic domain of ILK but not the N terminus or the PH domain (Supplementary Fig. 7a, b). To further investigate whether ILK is a substrate for HOIP, linear ubiquitination assays were performed. The results showed that LUBAC could conjugate linear ubiquitin chains to ILK, while LUBAC containing the HOIP catalytically inactive mutant (HOIP-C885S) could not (Fig. 4a and Supplementary Fig. 7d). ILK and the previously reported substrate ALK1 both belong to the serine-threonine protein kinase family[2,46]. This prompted us to compare their sequences to find the ubiquitination sites on ILK, which corresponded to K210 and K229, the identified linear ubiquitination sites on ALK1. The K209 site of ILK in the PH domain corresponded to the K210 site of ALK1, and K220 of ILK in the kinase catalytic domain corresponded to K229 of ALK1 (Supplementary Fig. 7e, f). Subsequent mutation assays showed that the linear ubiquitination of ILK-2KR (K209/K220R, in which lysine [K] residues in ILK were mutated to arginine [R]) was almost completely abolished compared with that of the wild-type ILK (Fig. 4b). Meanwhile, the immunoprecipitation assay confirmed that the ILK mutation (K209R/K220R/2KR) did not affect the interaction of ILK with HOIP (Supplementary Fig. 7g). These results showed that ILK could be linearly ubiquitinated at the K209 and K220 sites by LUBAC.

ILK combines the functions of a signal transductor and a scaffold protein through its interaction with integrins, facilitating further protein recruitment within the ILK–PINCH–Parvin complex (IPP)[46]. Therefore, we tested whether the formation of the IPP complex was affected by LUBAC. Immunoprecipitation assays showed that the

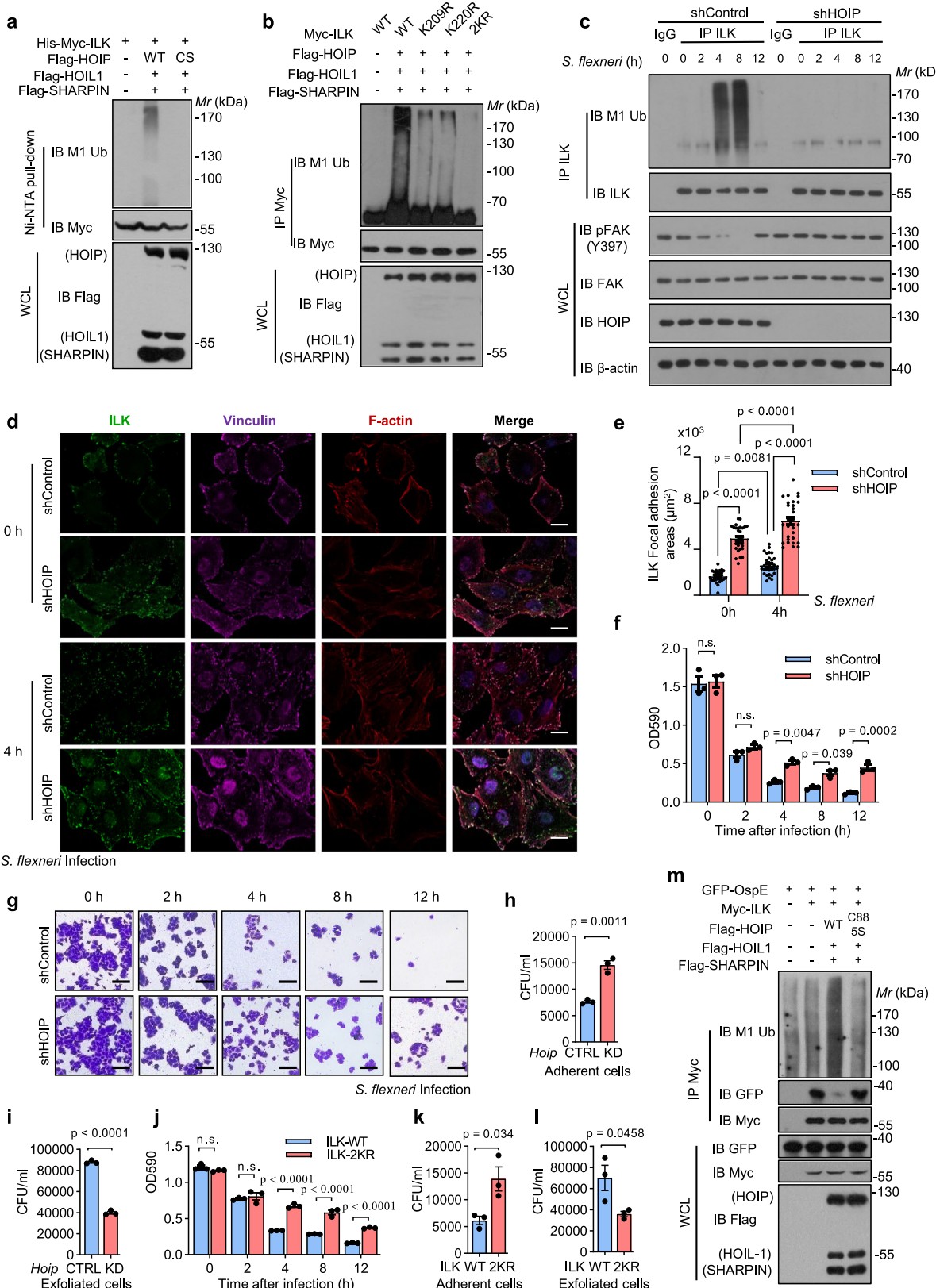

interaction between ILK and α-Parvin/β-Parvin remained unchanged when ILK was linearly ubiquitinated by LUBAC (Supplementary Fig. 7h, i), suggesting that the linear ubiquitination of ILK might regulate some specific conditions, such as *Shigella flexneri* infection, instead of normal physiological activities. Previous studies have shown that ILK and linear ubiquitination are crucial for autophagy regulation of bacteria[43,44,47]. However, the ILK inhibitor ILK-IN-3 enhanced the autophagy of *Shigella flexneri*-infected cells while HOIP inhibitor co-treatment failed to enhance the effect of ILK inhibitor (Supplementary Fig. 7j), suggesting that linear ubiquitination of ILK was not involved in autophagy. Emerging studies have shown that ILK participates in the infection processes of many pathogenic bacteria through the

**Fig. 4 | LUBAC provides defense against bacterial infection by linear ubiquitination of ILK. a** Ni-NTA pull-down of ILK linear ubiquitination in HEK293T cells transfected with Flag-tagged HOIP (or HOIP inactivated mutant), HOIL-1, SHARPIN and immunoblot with indicated antibodies. **b** Immunoprecipitation of ILK and mutants linear ubiquitination in HEK293T cells transfected with Myc-tagged ILK wild type (WT) or its mutants K209R, K220R or 2KR (K209R/K220R) and LUBAC, followed by immunoblotting with indicated antibodies. **c** Immunoprecipitation of ILK linear ubiquitination in control and HOIP knockdown Hela cells, infected with *Shigella flexner* for indicated time points. And immunoblot with indicated antibodies. **d** Control and HOIP knockdown HeLa cells were infected with *Shigella flexner* for indicated time points respectively, and then collected for immunofluorescence with indicated antibodies. Scale bars, 20 μm. **e** Quantification of ILK levels in focal adhesion areas in (**d**). n = 30 per group. **f, g** Representative images showing the attached cells of control and HOIP knockdown groups infected with *Shigella flexner* for indicated time points, Scale bars, 10 μm, and quantification of the OD590 of attached cells in (**f**). n = 3 per group. Quantification of bacterial load in the control and HOIP knockdown attached cells (**h**) and detached cells (**i**) after 12 h of *Shigella flexneri* infection. n = 3 per group. **j** Quantification of the $OD_{590}$ of attached cells control and ILK-2KR groups infected with *Shigella flexner* for the indicated time points. n = 3 per group. Quantification of bacterial load in the control and ILK-2KR attached cells (**k**) and detached cells (**l**) after 12 h of *Shigella flexneri* infection. n = 3 per group. **m** HeLa cells were transfected with GFP-tagged OspE, Myc-tagged ILK and Flag-tagged HOIP-WT, HOIP-C885S, HOIL-1, SHARPIN, and divided into two fractions: the one was used for co-immunoprecipitation assays and the other was used for ubiquitination assays, followed by immunoblotting indicated antibodies. Data are shown as the mean ± SEM; *p* values are from the unpaired two-sided *t*-test or two-way ANOVA (Sidak's multiple comparisons test). n.s., no significant. Source data are provided as a Source Data file.

regulation of focal adhesion (FA) turnover in the intestines[48–50]. Here, we found that the linear ubiquitination of ILK intensified from 0 h to 4 h, peaking at approximately 4 h, and was attenuated thereafter, receding to baseline levels at approximately 12 h (Supplementary Fig. 7k). Moreover, HOIP knockdown led to a lower level of linear ubiquitination of ILK induced by *Shigella flexneri* infection (Fig. 4c). The phosphorylation of FA kinase (FAK), which characterized FA disassembly, was also decreased in the HOIP knockdown group (Fig. 4c). The dynamic fluctuation in ILK linear ubiquitination was due to the dynamic interaction among HOIP, OTULIN and ILK in FAs during *Shigella flexneri* infection (Supplementary Fig. 7l, m). To further confirm the influence of LUBAC-mediated ILK linear ubiquitination on FA turnover after *Shigella flexner*i infection, the ILK localization in control and HOIP-knockdown HeLa cells with *Shigella flexneri* infection were detected. Immunofluorescence assays showed that the ILK levels in FA were increased in HOIP knockdown cells compared with control cells (Fig. 4d, e). Subsequent cell detachment assays also showed that more adherent cells were observed in the HOIP knockdown group at 4 h, 8 h, and 12 h after infection (Fig. 4f, g). Consistently, compared with control cells, the bacterial load in the HOIP-knockdown attached cells increased while the bacterial load in the HOIP-knockdown detached cells decreased (Fig. 4h, i). In addition, more adherent cells were observed in the ILK-2KR group at 4 h, 8 h, and 12 h after infection (Fig. 4j and Supplementary Fig. 7n). Meanwhile, compared with control cells, the bacterial load in the ILK-2KR attached cells increased while the bacterial load in the ILK-2KR detached cells decreased (Fig. 4k, l). All of these data suggested that linear ubiquitination of ILK promoted rapid FA turnover and increased cell detachment after *Shigella flexneri* infection. A canonical study has reported that *Shigella*'s toxic factor OspE hijacks ILK to stabilize FAs and block cell detachment[48]. Thus, we wondered whether the OspE-ILK axis was impacted by linear ubiquitination. The localization among linear ubiquitination, ILK and OspE was obviously observed in *Shigella flexneri* infected cells (Supplementary Fig. 7o). Moreover, the interaction between ILK and OspE decreased when ILK was modified by linear ubiquitination (Fig. 4m). Taken together, the above evidences indicated that LUBAC-mediated ILK linear ubiquitination might protect ILK from OspE hijacking by destroying the interaction between OspE and ILK, thus decreasing FA formation and promoting the detachment of infected cells.

### Linear ubiquitination of SMURF1 attenuates its E3 activity

Of the ubiquitin ligases we profiled, Smad ubiquitination regulatory factor 1 (SMURF1) caught our attention because it specifically interacted with HOIP in brain, liver, spleen, lungs and bones (Fig. 5a). Considering that the physiological role of linear ubiquitination catalyzed by HOIP has never been reported in bones, the possibility that HOIP functions in bones through SMURF1 was tested. We first tested the interaction between HOIP and SMURF1 in osteoblast cells. Endogenous SMURF1 was found to interact with HOIP in bone marrow mesenchymal stem cells (BMSCs) and 3T3-E1 cells (Fig. 5b, c).

Moreover, this interaction was confirmed by using an in vitro GST pull-down assay (Supplementary Fig. 8a). Domain mapping studies revealed that the WW domain of SMURF1 was responsible for the interaction with the HOIP LDD domain (Supplementary Fig. 8b–d). Subsequent linear ubiquitination analysis demonstrated that HOIP cooperated with HOIL-1 to ubiquitinate SMURF1, while the HOIP-CS mutant (C699,702, 871, 874 S, inactivated mutant) could not (Fig. 5d–f). Consistently, OTULIN cleaved the linear ubiquitin chains of SMURF1 catalyzed by LUBAC, while OTULIN C129A (catalytically inactive) failed (Supplementary Fig. 8e). SMAD1, SMAD5 and MEKK2 are the traditional substrates of SMURF1[51,52]. Knockdown of *Hoip* specifically decreased SMAD1/5 and MEKK2 protein levels while overexpression of HOIP increased SMAD1/5 and MEKK2 levels without affecting their mRNA levels (Fig. 5g, h and Supplementary Fig. 8f). These effects were probably because linear ubiquitination mediated the suppressed E3 ligase activity of SMURF1 toward these substrates. To further dissect the potential molecular mechanism, the dimerization of SMURF1, the interaction between SMURF1 and the ubiquitin, the interaction between SMURF1 and the substrate SMAD1, the interaction between SMURF1 and E2, or the linear ubiquitination of the substrates was assessed. The results showed that LUBAC conjugated linear ubiquitin chains onto SMURF1, but not SMADs (Supplementary Fig. 8g). Moreover, this linear ubiquitination of SMURF1 neither perturbed the dimerization of SMURF1 nor affected interactions of SMURF1 with the ubiquitin (or the substrates) (Supplementary Fig. 8h–j). However, the interaction between SMURF1 and E2 UBC12 (or UBCH5C) was significantly decreased due to LUBAC overexpression (Fig. 5i), suggesting that linear ubiquitination of SMURF1 attenuated its E3 activity by disrupting the E2-E3 interaction. This was also confirmed by the assessment of the linear ubiquitination sites of SMURF1. MS and mutation data showed that the K (lysine) 357, K624, K667, and K713 residues were conjugation sites that underwent LUBAC-mediated linear ubiquitination (Fig. 5j, k). The three-dimensional structure of the SMURF HECT domain (PDB: 1ZVD) further indicated that these ubiquitination sites were in the HECT domain, far away from the ATP binding site, perhaps disrupting the E2-E3 interaction, finally causing decreased degradative ubiquitination of SMURF1 substrates (Fig. 5j). Consistently, linear ubiquitination conjugated by LUBAC indeed decreased the ubiquitination level of SMURF1 substrate SMAD5, while SMURF1-4KR and HOIP inhibitor co-treatment failed to do so (Fig. 5l). Overall, these data indicated that linear ubiquitination led to the dampened E3 ligase activity of SMURF1 by disrupting the E2-E3 interaction.

### Deletion of Hoip causes a reduced bone mass phenotype

Previous studies have shown that SMURF1 is a negative regulator of osteogenesis in bones, which prompted us to determine the physiological role of HOIP in osteogenesis[52]. We first generated an osteoblast-specific *Hoip* deletion mouse model (Supplementary Fig. 9a–d). Compared with *Hoip*^loxP/wt^ *Osx-Cre*^+^ (referred to as CTRL)

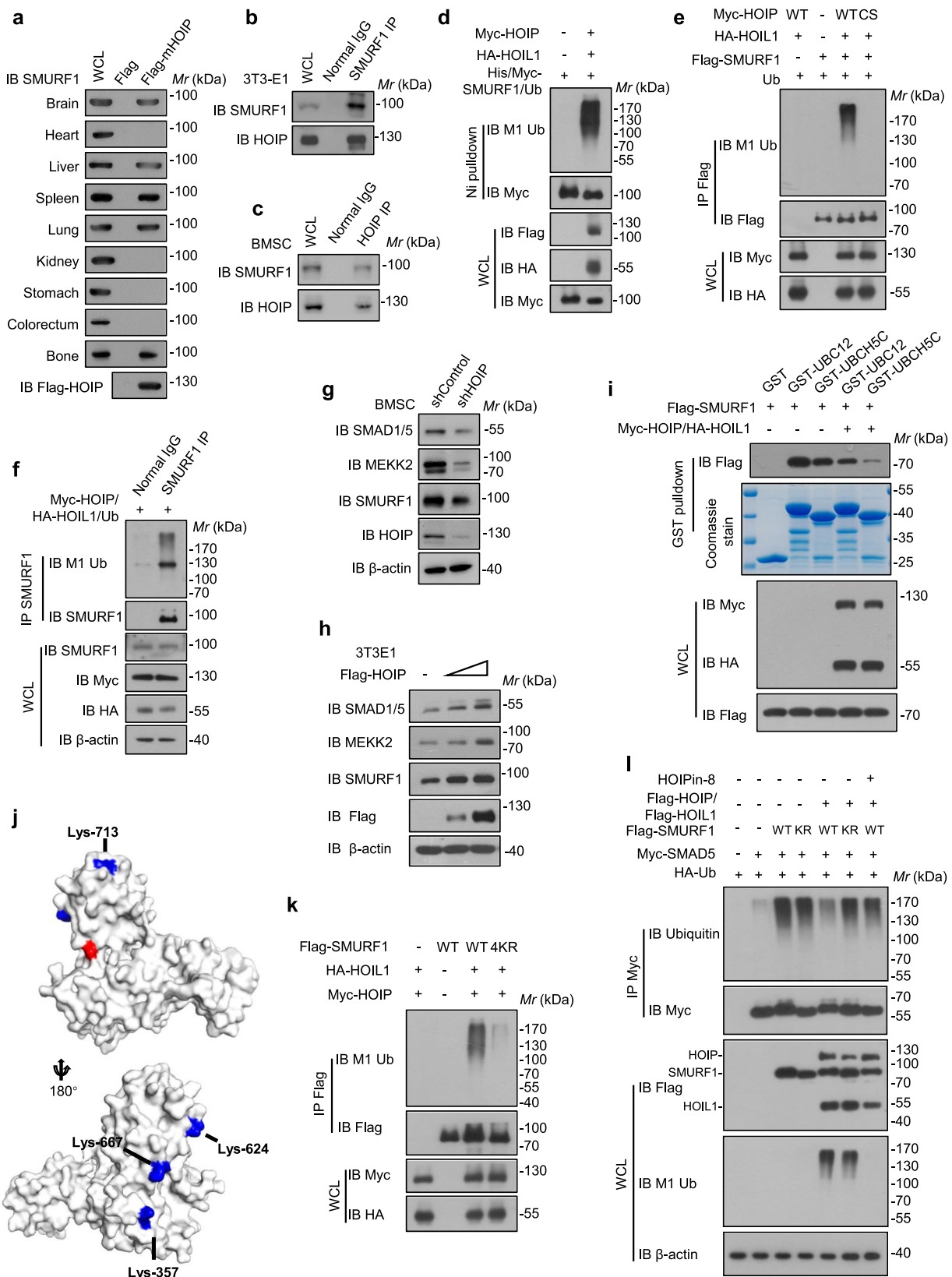

mice, *Hoip^{loxP/loxP} Osx-Cre^+* (referred to as CKO) mice developed marked dwarfism with decreased weight that was independent of sex (Fig. 6a, b and Supplementary Fig. 9e). Further biomechanical properties were analyzed by three-point bending, and the results showed that the maximum load and stiffness in the tibia bones of HOIP-conditional knockout (CKO) mice were markedly reduced

(Fig. 6c). ELISA of the bone formation marker N-terminal propeptide of type I procollagen (PINP) and the bone resorption marker C-terminal telopeptide of collagen type 1 (CTX-1) revealed a reduced bone formation rate in HOIP-CKO mice instead of bone resorption (Fig. 6d). Further microcomputed tomography (micro-CT) analysis of femurs from 6-, 12- and 18-week-old mice showed that HOIP-CKO

**Fig. 5 | Linear ubiquitination of SMURF1 attenuates its E3 activity.**
**a** Immunoprecipitation of ectopic expressed HOIP protein in nine tissues lysates and immunoblot with the SMURF1 antibody. Immunoprecipitates of SMURF1 in 3T3-E1 cells (**b**) and immunoprecipitates of HOIP in BMSC cells (**c**), followed by immunoblotting with the indicated antibodies. **d** Ni-NTA pull-down analysis of ILK linear ubiquitination in HEK293T cells transfected with LUBAC and immunoblot with the indicated antibodies. **e** Immunoprecipitates of SMURF1 linear ubiquitination in HEK293T cells transfected with LUBAC wildtype and inactive mutant. **f** Immunoprecipitates of endogenous SMURF1 ubiquitination in HEK293T cells transfected with LUBAC. **g** Immunoblot of SMURF1 and its substrate SMAD1/5 in HOIP knockdown BMSCs. **h** Immunoblot of SMURF1 and its substrate SMAD1/5, MEKK2 in BMSCs transfected with Flag-tagged HOIP. **i** Immunoblot analysis of the interaction between SMURF1 and its E2 UBCH5C, UBC12 in vitro. **j** Three-dimensional structure of the SMURF HECT domain (PDB: 1ZVD). Linear ubiquitin lysine sites in blue and catalytic center in red. **k** Immunoprecipitates of SMURF1 and its KR mutants' linear ubiquitination in HEK293T cells transfected with LUBAC. **l** Immunoprecipitates of SMAD1 ubiquitination in DMSO or HOIPin-8 treated HEK293T cells transfected with wildtype SMURF1, SMURF1 KR mutant or wildtype LUBAC. Source data are provided as a Source Data file.

mice began to develop marked bone mass loss after 6 weeks and maintained this severe phenotype until week 18 (Fig. 6e, f). Specifically, we found that 6-week-old HOIP-CKO mice were osteopenic, with reduced bone mineral density (BMD) and bone volume per tissue volume (BV/TV) in the femoral trabecular bones. Further analysis showed that the reduced trabecular number (Tb.N) of HOIP-CKO mice was accompanied by decreased trabecular thickness (Tb.Th) and an increased trabecular bone pattern factor (Tb.Pf) compared with those of control mice. Additionally, Subsequent von Kossa staining, Safranin O staining and TRAP staining further confirmed that HOIP CKO decreased bone mineral deposition without affecting chondrocytogenesis or osteoclastogenesis in HOIP-CKO mice (Fig. 6g–j). Consistently, double calcein labeling showed a decreased bone formation rate in HOIP-CKO mice compared with their littermate controls (Fig. 6k). Histomorphometric analysis also revealed a consistent, significant reduction in osteoblast numbers (N.Ob/B. Pm (/mm)) and osteoblast surface area (Ob. S/BS (%)) (Fig. 6l, m).

To further confirm the positive role of HOIP in osteogenesis, we isolated bone marrow mesenchymal stem cells (BMSCs) from femurs to evaluate the role of HOIP in osteoblast proliferation, differentiation and mineralization. Our data showed that HOIP deficiency significantly affected the formation of fibroblast colony-forming unit (CFU-F) and osteoblast colony-forming unit (CFU-Ob) colonies (Fig. 7a). Moreover, HOIP deletion markedly suppressed osteoblast differentiation and mineralization (measured by alkaline phosphatase [ALP] staining for the differentiation assay, as well as Alizarin red staining for the mineralization assay) (Fig. 7b). Consistently, the expression of osteoblast differentiation markers, such as *Col1a1*, *Osx*, *Runx2* and *Ocn*, decreased in HOIP-deficient BMSCs (Fig. 7c). More importantly, immunohistochemical staining of linear ubiquitination, RUNX2 (the osteoblast cell marker) and SMURF1 (or SMAD1/5) showed that the signal of RUNX2 positive osteoblasts decreased in femurs from HOIP CKO mice and the colocalization between linear ubiquitination and SMURF1 (or SMAD1/5) in osteoblasts also decreased in these HOIP CKO femurs (Fig. 7d, e). Consistently, deletion of *Hoip* specifically decreased the ubiquitination of SMURF1 and promoted the degradation of SMURF1 substrates (such as SMAD1/5 and MEKK2) without affecting the TGFβ signaling (Fig. 7f, g). Similar results were observed in BMSCs treated with the HOIP inhibitor[53]. HOIP inhibitor inhibited the linear ubiquitination of Smurf1 and decreased the proliferation, differentiation and mineralization of BMSCs (Supplementary Fig. 9f–i). The expression of the SMURF1 substrate SMAD1 and osteoblast differentiation markers (such as *Alp*, *Ocn*, *Osx*) also decreased with the HOIP inhibitor treatment (Supplementary Fig. 9j, k). All of these data suggested that linear ubiquitination of SMURF1 decreased its own E3 ligases activity in vivo. Therefore, BMSCs derived from HOIP CKO femurs were treated with the Smurf1 inhibitor A17[54] to test whether the E3 ligase activity of SMURF1 was required for bone phenotypes in HOIP CKO mice. The results showed that the mRNA levels of osteogenesis-related markers (such as *Alp*, *Col1a1* and *Osx*) and protein levels of SUMRF1 substrates (such as SMAD1/5, MEEK2) were both enhanced in HOIP CKO BMSCs, when compared to untreated groups (Fig. 7h and Supplementary Fig. 9l). ALP staining of CKO BMSCs treated with A17

also confirmed the enhanced osteogenesis when compared with untreated groups (Fig. 7i), which further confirmed that HOIP promoted osteogenesis by antagonizing Smurf1 enzyme activity. Altogether, these data showed that HOIP deficiency led to reduced bone mass phenotypes depending on Smurf1, suggesting an undefined physiological role of HOIP-mediated linear ubiquitination in bone development.

## Discussion

Our proteomic analyses depict a HOIP interactome with unprecedented spatial resolution across tissues (Fig. 8). Overall, we quantitatively analyze the four HOIP domain-specific interactomes in nine different normal mouse tissues. Proteins that are enriched in a single tissue or in most tissues are identified and analyzed regarding to their biological functions and roles in genetic diseases. Our platform also identifies a group of HOIP domain-enriched proteins that have not been previously identified. Furthermore, our HOIP interactome provides an unified resource that will aid in further elucidation of the pivotal roles of HOIP-mediated linear ubiquitination in *Shigella flexneri* infection and osteogenesis.

Most previous studies have focused on the functional elucidation of linear ubiquitination in a single organ, thereby limiting the generalization and comparison of the findings to many other tissues simultaneously[9,11–16]. Our HOIP PPI analysis provided systematic insights into key roles of linear ubiquitination in physiological and pathological processes across tissues. Essential biological processes, such as chromatin organization, endocytosis, cellular respiration and small molecule catabolic processes, were enriched in most tissues. This suggested that HOIP-mediated linear ubiquitination was indispensable for essential cellular functions. Moreover, arrays of organ-specific HOIP PPIs were detected, for example, brain-specific PPIs HAP1 and SYN1, heart-specific PPIs MYL3 and CSRP3, liver-specific PPIs SCP2 and PRKAA2, spleen-specific PPIs TRAF1 and TRAF2, and colorectal-specific PPIs CDHR5 and ABCB1, providing a systematic understanding of linear ubiquitination functions in every tissue. Defects in linear ubiquitination signaling are associated with immune disorders, cancers and neurogenerative diseases[1]. In OMIM analysis of HOIP PPIs, tissue-specific disease symptoms, such as cortical dysplasia, cardiomyopathy and polycystic kidney disease, suggested the presence of previously unappreciated functions or regulatory aspects of linear ubiquitination. Moreover, the multitissue disease symptom, such as mitochondrial complex I deficiency, was highly enriched in the heart, stomach and bones, providing an in-depth view of the dependence of complicated diseases on the common regulatory mechanism of linear ubiquitination.

A few reports have indicated that mutation or truncation of these domains can affect certain functions of LUBAC[25,26], but there is no systematic comparative assessment of the contributions of individual HOIP domain to LUBAC functions. In this study, we systematically analyzed HOIP domain-mediated functions. As expected, we found that some undefined processes, such as RNA splicing and metabolic processes, were coregulated by the PUB domain, NZF domain and RBR domain based on the co-enrichment of these terms in multiple domains. However, domain-specific enrichment analysis provided

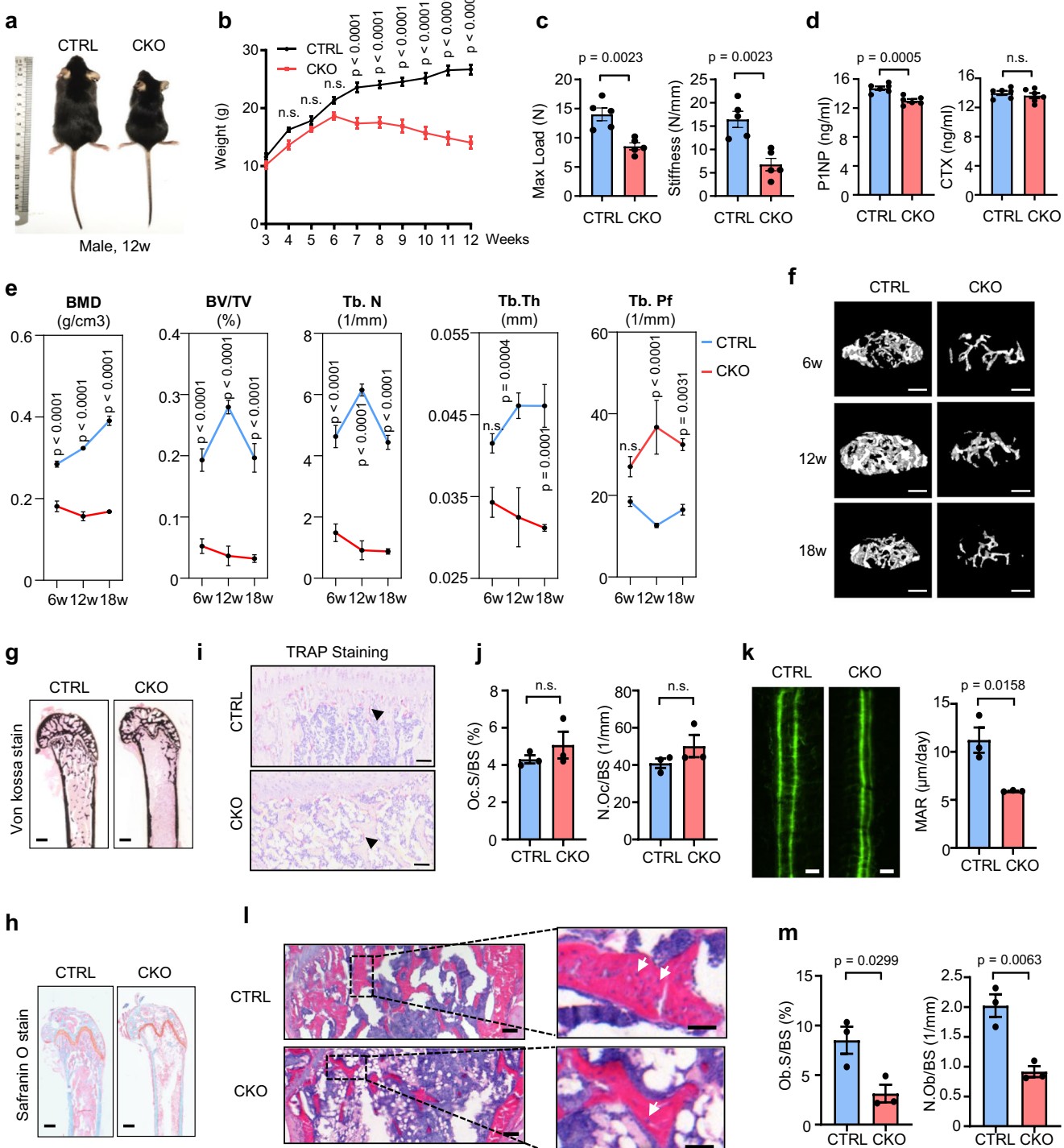

**Fig. 6 | Deletion of *Hoip* causes a reduced bone mass phenotype.**
**a** Representative image of 8-week-old male *Hoip^loxP/wt Osx-Cre^+* (referred to as CTRL) *Hoip^loxP/loxP Osx-Cre^+* (referred to as CKO) mice. **b** Weight monitoring of HOIP CTRL and CKO mice. *n* = 5 per group. **c** Quantification of maximal loading and stiffness of humeral diaphysis from 8-week-old HOIP CTRL and CKO mice. *n* = 5 per group. **d** Serum levels of N-terminal propeptide of type I procollagen (PINP) and C-terminal telopeptide of collagen type 1 (CTX-1) from 8-week-old HOIP CTRL and CKO mice. *n* = 6 per group. **e** Histomorphometric analysis of trabecular bones from 6, 12,18-week-old HOIP CTRL and CKO mice, including bone mass density (BMD), bone volume per tissue volume (BV/TV), trabecular thickness (Tb.Th), trabecular number (Tb.N), trabecular spacing (Tb.Sp). *n* = 5 per group. **f** Representative micro-CT images of trabecular bones from 6, 12,18-week-old HOIP CTRL and CKO mice. Scale bars, 0.1 cm. **g** Von Kossa staining of femurs from 8-week-old HOIP CTRL and CKO mice. Scale bars, 0.2 mm. **h** Safranin O staining of femurs from 8-week-old HOIP

CTRL and CKO mice. Scale bars, 0.2 mm. TRAP staining (**i**) of femurs from 8-week-old HOIP CTRL and CKO mice and quantification (**j**) of osteoclast surface/ millimeter of bone perimeter (Oc.BS) and osteoclast number/ millimeter of bone perimeter (N.Oc/BS) in (**g**). *n* = 3 per group, Scale bars, 50 μm. **k** Representative images of calcein double staining and quantitative analysis of mineralization apposition rate (MAR) of femurs from 8-week-old HOIP CTRL and CKO mice. *n* = 3 per group, Scale bars, 100 μm. **l** H&E staining of femurs from 8-week-old HOIP CTRL and CKO mice. Scale bars, 200 μm. **m** Quantitative analysis of osteoblast number/ millimeter of bone perimeter (N.Ob/BS) and osteoblast surface/ millimeter of bone perimeter (Ob.S/BS) in femurs from (**l**). *n* = 3 per group. All the data are shown as the mean ± SEM; *p* values are from the unpaired two-sided *t*-test and two-way ANOVA (Sidak's multiple comparisons test). n.s., no significant. Source data are provided as a Source Data file.

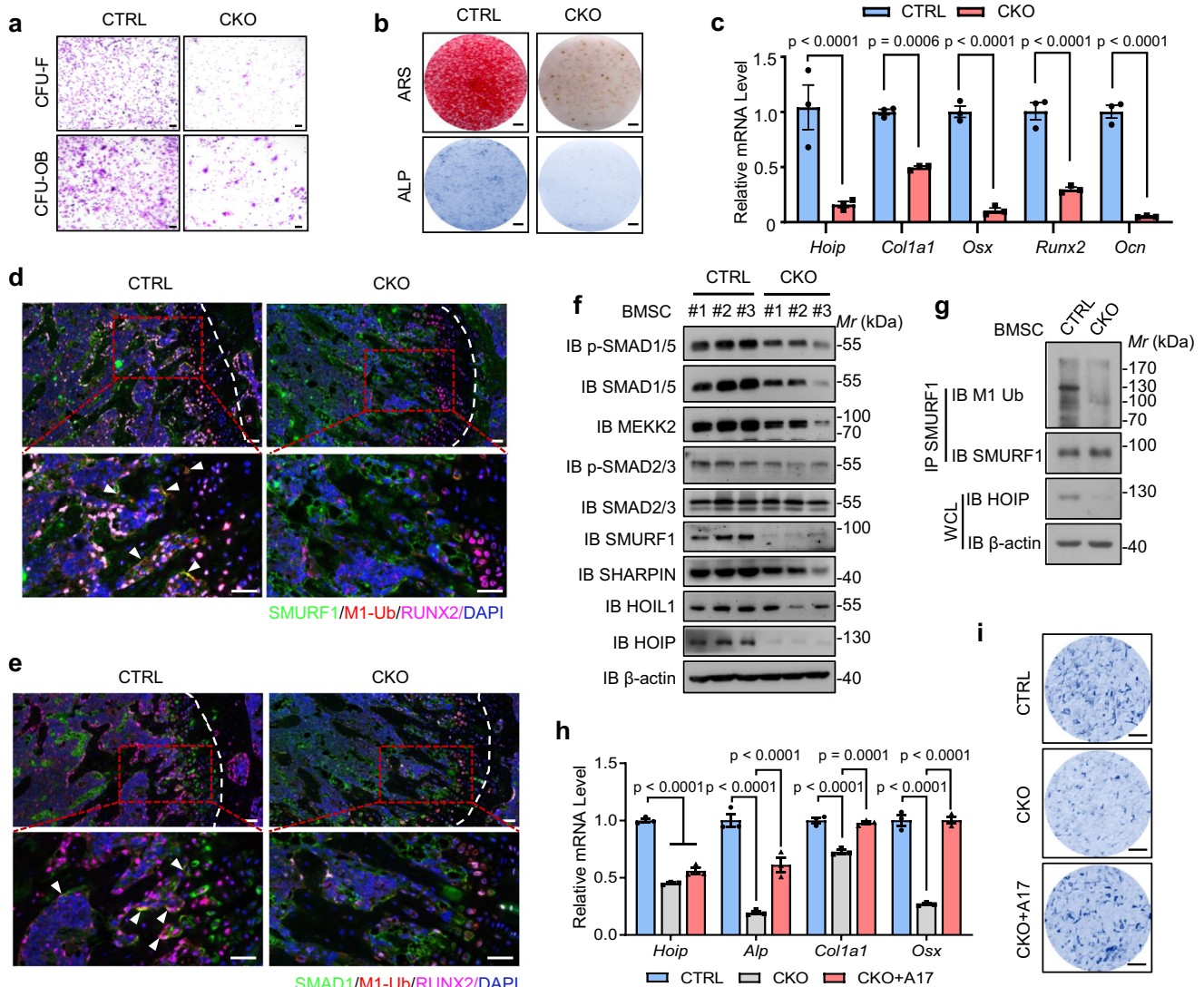

**Fig. 7 | SMURF1 inhibitor rescues the osteogenesis defects in HOIP knockout cells. a** Representative image of fibroblast colony-forming unit (CFU-F) and osteoblast colony-forming unit (CFU-Ob) of BMSCs from *Hoip^loxP/wt Osx-Cre^+* (referred to as CTRL) *Hoip^loxP/loxP Osx-Cre^+* (referred to as CKO) mice. Scale bars, 50 μm. **b** Representative images of alkaline phosphatase (ALP) staining and Alizarin red staining (ARS) of bone marrow mesenchymal stem cells (BMSCs) from HOIP CTRL and CKO mice after cultured in osteogenic medium for 14 and 28 days. Scale bars, 2 mm. **c** Quantitative RT-PCR analysis of osteogenesis genes mRNA levels in BMSCs from 8-week-old HOIP CTRL and CKO mice. *n* = 3 per group. **d** Immunohistochemical staining of linear ubiquitination, RUNX2 (the osteoblast cell marker) and SMURF1 in femurs from control and HOIP CKO mice. Scale bars,

50 μm. **e** Immunohistochemical staining of linear ubiquitination, RUNX2 and SMAD1 in femurs from control and HOIP CKO mice. Scale bars, 50 μm. **f** Immunoblot of SMURF1 and its substrates SMAD1/5, MEKK2 in BMSCs from HOIP CTRL and CKO mice. **g** Immunoprecipitates of SMURF1 linear ubiquitination in BMSCs from HOIP CTRL and CKO mice. **h** Quantitative RT-PCR analysis of osteogenesis genes mRNA levels in BMSCs from CTRL, CKO and CKO treated with SMURF1 inhibitor A17 groups for 28 days. *n* = 3 per group. **i** Representative images of ALP staining of BMSCs from CTRL, CKO and CKO treated with SMURF1 inhibitor A17 groups for 14 days. Scale bars, 0.5 mm. All the data are shown as the mean ± SEM; *p* values are from the two-way ANOVA (Sidak's multiple comparisons test). Source data are provided as a Source Data file.

more information. HOIP participates in NF-κB signaling, cell death, inflammation, immunity, and cancer by conjugating linear ubiquitin chains onto a few proteins, such as RIPK1, RIPK2 and NEMO[4–7]. We characterized potential linearly ubiquitinated substrates recognized by the NZF domain that were involved in glycolytic processes, RNA splicing and oxidative stress that have not previously been reported, expanding the potential substrates of linear ubiquitination. Further analysis showed the coordination of OTULIN with the HOIP PUB domain in mitotic spindle organization and the catalytic activity of the HOIP RBR domain in autophagy. Although LUBAC has previously been shown to regulate chromosome alignment in mitosis[42] and autophagy initiation and maturation[43,44], these HOIP domain-mediated interactions provide an additional perspective for understanding HOIP functions. Moreover, the unexpected functions of the HOIP UBA

domain in protein localization might suggest a regulatory mechanism for the cellular localization of HOIP, which warrants further study. Additionally, PPIs of HOIP NZF domain were found to be specific in liver, brain, and colorectum when compared to the other domains. This was probably due to tissue-specific post-translational modifications in specific domains or the tissue-specific expression of HOIP interactors.

We further investigated the functions of HOIP mediated linear ubiquitination across tissues and discovered two undefined functions of linear ubiquitination in intestinal infection and bone development. Kinase analysis of HOIP PPIs identified ILK, an important cytoplasmic component of cell-ECM adhesions, as a potential substrate of HOIP. Subsequent experiments showed that linear ubiquitination of ILK might protect ILK from hijacking by *Shigella*'s toxic

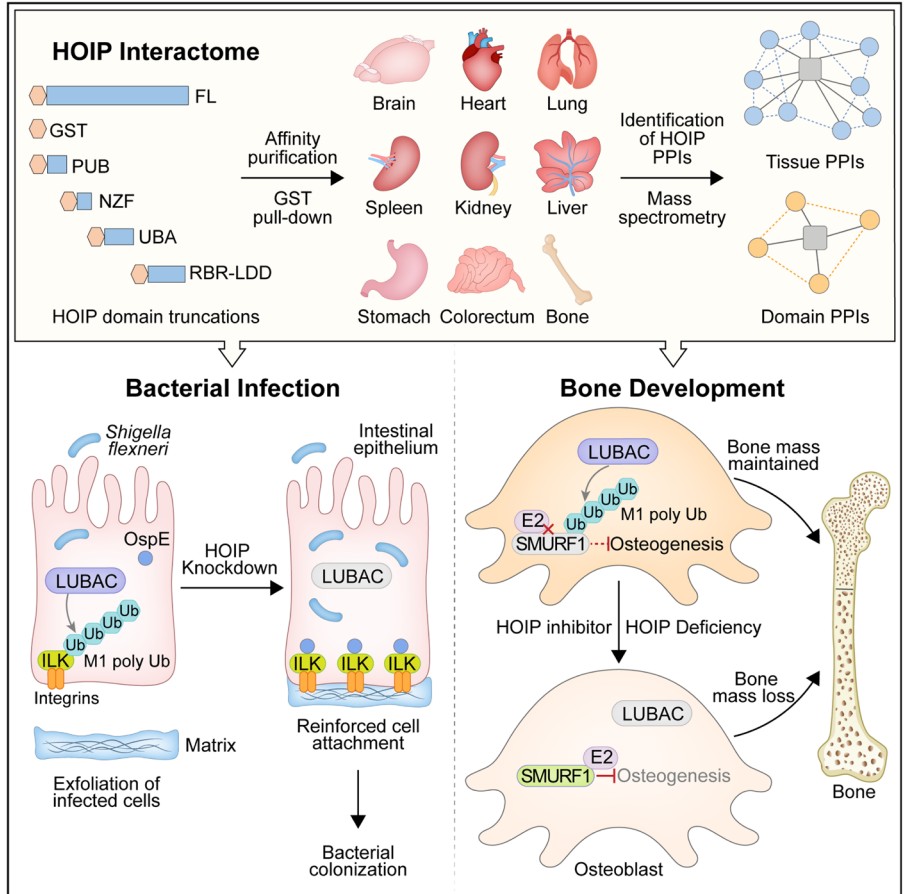

**Fig. 8 | The proposed working model.** Top, HOIP domain- and tissue- interactome profiling indicates crucial regulatory functions and elaborate signaling networks of linear ubiquitination. Bottom-left, HOIP-mediated ILK linear ubiquitination promotes cell defense against *Shigella flexneri* infections. Bottom-right, HOIP-mediated SMURF1 linear ubiquitination antagonizes SMURF1-E2 interaction and thus promoting osteogenesis.

factor OspE through destroying the interaction between OspE and ILK, thus promoting the detachment of infected cells. Previously, *Shigella* ubiquitin ligases IpaH1.4 and IpaH2.5 have been found to modify HOIP with Lys48-Ub and targeted the modified HOIP for degradation[17]. Here, we revealed a previously undefined strategy employed by the host to avoid infections, providing more support for linear ubiquitination in the bacterial immune response. E3 ligase analysis of HOIP PPIs showed that SMURF1, a negative regulator of osteogenesis, was regulated by linear ubiquitination. Linear ubiquitination led to dampening of the E3-ligase activity of SMURF1 by disrupting the E2-E3 interaction. This reveals a different negative regulatory mechanism of SMURF1 that differs from SMURF1 activation via CKIP-1[55] and SMURF1 degradation by FBXL15[56]. Interestingly, our study also demonstrated that *Hoip* deficiency in osteoblasts caused a reduction in bone mass and that this defect was caused by reduced linear ubiquitination of SMURF1. Although some articles have identified functions of HOIP mediated linear ubiquitination in Treg cells, B cells, liver parenchymal cells, macrophages, epithelial cells and endothelial cells[9,11–16], our report describes a undefined role of linear ubiquitination in osteoblast cells.

In summary, the biological relevance of HOIP in *Shigella flexneri* infection and osteogenesis, suggests the indispensable roles of linear ubiquitination in physiological and pathological processes. Moreover, our systematically constructed HOIP PPIs across domains and tissues provide an outstanding paradigm for a more comprehensive understanding of HOIP and an important framework for further functional elucidation of linear ubiquitin chains, a functionally important atypical type of ubiquitin linkage.

## Methods

### Ethics
The Institutional Animal Care and Use Committee of the Beijing Institute of Lifeomics is responsible for ethical compliance approval of all animal protocols (IACUC-DWZX-2023-P503).

### Mouse strains
All mice were on a C57BL/6 background and maintained under specific pathogen-free conditions in individually ventilated cages (12 h light/dark cycle, 50% relative humidity, between 25 and 27 °C). 8-week-old *Hoip*^loxP/loxP mice were obtained from RIKEN (cat# RBRC09483). 8-week-old wild-type mice and transgenic mice expressing *Cre* recombinase under control of the *Osx* (also known as *Sp7*) promoter (C57BL/6-*Sp7*^tm1(icre)/Bcgen) were purchased from BIOCYTOGEN (cat# 110131). *Hoip*^loxP/loxP mice (approximately 8–12 weeks) and *Osx*-Cre mice were used to generate *Hoip* osteoblast-specific deletion mice (*Hoip*^loxP/loxP *Osx-Cre*^+).

### Cell culture
The HEK293T (ATCC, CRL-3216) and Hela cells (ATCC, CCL-2) were cultured in Dulbecco's modified Eagle's medium (DMEM, Gibco, C11995500BT-1) supplemented with 10% fetal bovine serum (FBS, Cellmax, SA301.02.V). The 3T3-E1 cells (ATCC, CRL-2594) were maintained in α-MEM (Gibco, C12571500BT) with 10% FBS (Gemini, 900-108). BMSC were isolated from femurs of 8-week-old male mice and maintained in α-MEM (Gibco, C12571500BT) supplemented with 15% (v/v) FBS (Gemini, 900-108) and 1% (v/v) penicillin/streptomycin (CellWorld, C0160-611).

## GST pull-down and sample preparation for mass spectrometry

Tissues, including brain, heart, liver, spleen, lungs, kidneys, stomach, colorectum and bones were obtained from 8-week-old wild-type mice and ground to powder in liquid nitrogen, followed by the lysis in HEPES lysis buffer (20 mM HEPES (pH 7.2), 150 mM NaCl, 0.5% Triton X-100, 1 mM NaF and 1 mM dithiothreitol). GST only, GST-tagged HOIP PUB, NZF, UBA, RBR-LDD were purified from bacteria BL21 and bound to GST-tag Purification Resin (Beyotime, P2251), followed by incubation in tissue lysates at 4 °C overnight. After washing four times with HEPES lysis buffer, the resin was boiled for the immunoblot and mass spectrometry.

## Mass spectrometry

The bound proteins were separated by SDS-PAGE, and the gels were stained with Coomassie Blue. Bands representing putative interacting proteins were cut out and subjected to in-gel trypsin digestion with sequencing grade modified trypsin. The resulting peptide mixtures were analyzed by the LC-MS/MS system that consisted of an online Easy-nLC 1200 nano-HPLC system (Thermo Fisher Scientific) and the Orbitrap Fusion lumos mass spectrometer (Thermo Fisher Scientific). In brief, 0.5 μg of peptide mixture resolved in buffer A (0.1% formic acid (FA)) were loaded onto a 1 cm self-packed trap column (150 μm inner diameter, ReproSil-Pur C18-AQ, 3 μm resin; Dr Maisch) using buffer A and separated on a 150 μm inner diameter column with a length of 15 cm (ReproSil-Pur C18-AQ, 1.9 μm resin; Dr Maisch) over a 78 min gradient (buffer A, 0.1% FA in water; buffer B, 0.1% FA in ACN/water(80:20)) at a flow rate of 600 nl/min (0–8 min, 6–12% B; 8–58 min,12–30% B; 58–70 min, 30–40% B; 70–71 min, 40–95% B; and 71–78 min, 95% B). The spectrometer was operated in the positive-ion mode at an ion transfer tube temperature of 320 °C. The positive-ion spray voltage was 2.0 kV. For a full mass spectrometry survey scan, the scan ranged from 300 to 1400 m/z at a resolution of 120,000 and a maximum injection time of 50 ms. The MS2 spectra were acquired in the ion trap in rapid mode with an AGC target of 5000 and a maximum injection time was set to dynamic, and the dynamic exclusion duration was set to 20 s. The acquired MS/MS spectra were searched against the mouse Uniprot (updated on 20220113; 17,067 protein entries) with maxquant (2.0.3.0). The mass tolerances were 20 ppm for precursor ions and 0.5 Da for productions. Peptides and proteins were filtered with a false discovery rate of 1% using the decoy database strategy.

## Data analysis

Differential expressed proteins were obtained by using R packages, ProstaR and DAPAR (1.30.7). For each condition of the label-free quantification data, we required a protein to have non-zero iBAQ in two out of the three replicates. Then the iBAQ was logarithmic transformed and normalized by median centering. The missing values were replaced with 1% quantile. Differentially expressed proteins were detected by using a Limma moderated t-test. We used BioGRID[57] curated HOIP interactions as positive dataset. Our data contained 147 known interactions (41%), and we found that the foldchange of 2 maximizes all known interactions. Proteins were regarded as interactions between four domains and GST only group with P-value less than 0.05, and fold change cut-off was set as 2. The functional classification of differentially expressed proteins with Gene Ontology was carried out by an R package clusterProfiler (4.6.2). GO terms with Q-values less than 0.05 were considered to be significantly enriched. The network of human proximal proteins was visualized by Cytoscape (v3.9.1). The conversion of human and mouse homologous genes is performed using the R package homologene. A Perl script is used to calculate and count the number of interactions identified by different tissues and domains, as well as the Jaccard index for different tissues and domains.

## Evaluation of protein-protein interactions (PPIs)

To evaluate the quality of our data, we cross-referenced published human interaction omics data including BioPlex2.0[58], BioPlex3.0[59], HI-II-14[59], HURI[60], QUBIC[61] and literature curated dataset Lit-BM8 against the recall of BioGRID databases. To investigate the effect of such biases on the current coverage of the protein interactome network, we organized the HOIP protein-protein interactions by ranking proteins PubMed publications.

## Quantifying coverage of protein families

Several data sources were used to determine the fractions of various protein families including kinases, transcription factors, E3 ligases and deubiquitin enzymes (DUB). Kinase information is derived in Uniprot pkinfam (release 2020_04 of 12-Aug-2020). Transcription factors of mouse was downloaded in AnimalTFDB3.0 (http://bioinfo.life.hust.edu.cn/AnimalTFDB). E3 and DUB information are derived in IUCCD (https://gps.biocuckoo.org).

## Tissue enriched proteins and genetic diseases analysis

To identify the link between HOIP PPIs in tissues and particular genetic diseases, we compared the frequency of specific disease terms associated with HOIP PPIs in each tissue to the frequency of these terms in the background set (HOIP PPIs presented in all nine tissues) after excluding PPIs identified in corresponding tissues. Fisher's exact test was applied to assess the significance of the overlap.

## Immunoblotting

Both lysates and immunoprecipitants were examined using the indicated primary antibodies against GAPDH (Santa Cruz, Cat# sc-365062, RRID:AB_10847862, 1:2000 for IB), β-actin (Proteintech, Cat# 66009-1-Ig, RRID:AB_2687938, 1:4000 for IB), Myc (MBL, Cat# M047-3, RRID:AB_591112, 1:4000 for IB, 1:500 for IP), Flag (MBL, Cat# M185-3, RRID:AB_10950447, 1:1000 for IB, 1:500 for IP), HA (MBL, Cat# M180-3, RRID:AB_10951811, 1:1000 for IB), HOIP (Abcam, Cat# ab46322, RRID:AB_945269, 1:300 for IB), HOIP (Abcam, Cat# ab125189, RRID:AB_10976137, 1:500 for IB, 1:200 for IP), HOIL-1 (Sigma-Aldrich Cat# HPA024185, RRID:AB_1845673, 1:300 for IB), SHARPIN (Proteintech, Cat# 14626-1-AP, RRID:AB_2187734, 1:1000 for IB), OTULIN (Cell Signaling, Cat# 14127, RRID:AB_2576213, 1:1000 for IB), SMURF1(Abcam, Cat# ab57573, RRID:AB_945548, 1:500 for IB), SMURF1 (Abnova, Cat# H00057154-M01, RRID:AB_566195, 1:500 for IB, 1:200 for IP), SMAD1/5 (Abcam, Cat# ab75273, RRID:AB_1310686, 1:500 for IB), SMAD1(Santa Cruz Biotechnology, Cat# sc-7965, RRID:AB_628261, 1:400 for ICC), MEKK2 (Proteintech, Cat# 55106-1-AP, RRID:AB_11064604, 1:1000 for IB), AP2B1 (Proteintech, Cat# 15690-1-AP, RRID:AB_2056351, 1:1000 for IB), AP2M1 (Proteintech, Cat# 27355-1-AP, RRID:AB_2880853, 1:1000 for IB), RUVBL1 (Proteintech, Cat# 10210-2-AP, RRID:AB_2184405, 1:1500 for IB), RUVBL2 (Proteintech, Cat# 10195-1-AP, RRID:AB_2184679, 1:1500 for IB), SYN1 (Proteintech, Cat# 20258-1-AP, RRID:AB_2800493, 1:1500 for IB), HAP1 (Proteintech, Cat# 25133-1-AP, RRID:AB_2879915,, 1:500 for IB), MYL3 (Proteintech, Cat# 10913-1-AP, RRID:AB_2147607, 1:800 for IB), PRKAA2 (Proteintech, Cat# 18167-1-AP, RRID:AB_10695046, 1:1000 for IB), TRAF1(Proteintech, Cat# 26845-1-AP, RRID:AB_2880655, 1:1500 for IB), TRAF2 (Proteintech, Cat# 26846-1-AP, RRID:AB_2880656, 1:1500 for IB), CDHR5 (Proteintech, Cat# 25619-1-AP, RRID:AB_2880164, 1:1000 for IB), CSRP3 (Proteintech, Cat# 10721-1-AP, RRID:AB_2292475, 1:1000 for IB), ABCB1 (Proteintech, Cat# 22336-1-AP, RRID:AB_2833023, 1:2000 for IB), SCP2 (Proteintech, Cat# 23006-1-AP, RRID: AB_2879197, 1:1000 for IB), STAT1 (Proteintech, Cat# 10144-2-AP, RRID:AB_2286875, 1:3000 for IB, 1:500 for IP), β-Catenin (Abcam, Cat# ab32572, RRID:AB_725966, 1:1000 for IB, 1:50 for IP), FXR1 (Proteintech, Cat#13194-1-AP, RRID:AB_2110702, 1:1000 for IB), Ubiquitin (Cell Signaling Technology Cat# 20326, RRID:AB_3064918, 1:1000 for IB), M1 Ub (Lifesensors, Cat# AB130, RRI-

D:AB_2576211, 1:300 for IB, 1:100 for IHC), GFP (Proteintech, Cat# 50430-2-AP, RRID:AB_11042881, 1:1000 for IB, 1:1000 for IF), ILK (Proteintech, Cat# 12955-1-AP, RRID:AB_2127053, 1:200 for IB), ILK (Proteintech, Cat# 67724-1-Ig, RRID:AB_2882910, 1:500 for IB, 1:50 for IP), ILK ((Santa Cruz Biotechnology, Cat# sc-137221, RRID:AB_2127074, 1:500 for IB, 1:50 for IF), αTubulin (Biodragon, Cat# B1052, RRID:AB_2936302, 1:1000 for IB), FAK (Abclonal, Cat# A11131, RRID:AB_2758423, 1:1000 for IB), FAK (Y397) (Abclonal, Cat# AP0302, RRID:AB_2771470, 1:500 for IB), Vinculin (Proteintech, Cat# 26520-1-AP, RRID:AB_2868558, 1:300 for IF), α-parvin (Proteintech, Cat# 55268-1-AP; RRID:AB_10951112, 1:500 for IB); Goat anti-Rabbit IgG (H + L) Secondary Antibody (Thermo Fisher Scientific, Cat# 65-6120, RRID:AB_2533967, 1:2500 for IB, 1:1000 for IHC); Goat anti-Mouse IgG (H + L) Secondary Antibody, HRP (Thermo Fisher Scientific, Cat# 31430, RRID:AB_228307, 1:2500 for IB, 1:1000 for IHC); Alexa Fluor 488 Recombinant Polyclonal Antibody (Thermo Fisher Scientific, Cat# 710369, RRID:AB_2532697, 1:1000 for IF); Goat anti-Rabbit IgG (H + L) Cross-Adsorbed Secondary Antibody, Alexa Fluor™ 594 (Thermo Fisher Scientific, Cat# A-11012, RRID:AB_2534079, 1:1000 for IF); Donkey anti-Rabbit IgG (H + L) Highly Cross-Adsorbed Secondary Antibody, Alexa Fluor™ 647 (Thermo Fisher Scientific, Cat# A-31573, RRID:AB_2536183 1:1000 for IF); DAPI (Thermo Fisher Scientific, 1:1000 for IF) and detection was done by the SuperSignal™ West Pico PLUS chemiluminescent Detection Reagent (Thermo).

### Plasmids, cell transfection and immunoprecipitation

The plasmid information of HOIP, HOIL-1, SHARPIN, SMURF1, SMADs and related mutants were described in the previous studies[2,62]. Potential HOIP-interacting proteins, such as HK1, HK2 were cloned into the 3 × Flag-CMV14 vectors. ILK and its mutants were cloned into the pcDNA 3.1 vectors. HEK293T were transfected with these indicated plasmids using StarFect Transfection Reagent (GenStar, C101-10) according to the protocol. Cells were lysed with HEPES lysis buffer with the protease inhibitor (MedCHemExpress, HY-K0010) and PhosSTOP (Solarbio, P1260). Immunoprecipitation was performed using the indicated primary antibody and protein A/G agarose beads (Santa Cruz) at 4 °C, which were then washed with HEPES buffer three times. The lysates and immunoprecipitates were analyzed by immunoblotting.

### Ni-NTA pull-down

HEK293 cells were transfected with indicated His-tagged plasmids for 48 h and lysed in denaturing condition (buffer 1: 6 M guanidine-HCl, 0.1 M $Na_2HPO_4$/$Na_2H_2PO_4$, 10 mM imidazole (pH 8.0)). After sonication and isolation, the lysate incubated with nickel-nitrilotriacetic acid (Ni-NTA) agaroses (QIAGEN, 30210) for 2 h at room temperature. These pull-down agaroses were washed sequentially once in buffer 1, twice in buffer 1/2 mixture (buffer 1:buffer 2 = 1:3), and once in buffer 2 (25 mM Tris-HCl and 20 mM imidazole (pH 6.8)). The lysates and immunoprecipitates were analyzed by immunoblotting.

### Ubiquitylation analysis

Cells transfected with indicated plasmids were pelleted with 1% SDS and boiled in 100 °C for 10 min. These samples were then diluted in TNE buffer (50 mM Tris (pH 7.5), 150 mM NaCl, 1% NP-40, 1% Triton X-100, 0.5% sodium deoxycholate, 10 mM NaF, 1 mM $Na_3VO_4$ and proteinase inhibitor (Selleck)) with 0.1% SDS for subsequent immunoprecipitation. The lysates and immunoprecipitates were analyzed by immunoblotting.

### In vitro osteoblastic differentiation and mineralization

Primary BMSCs from 8-week-old male mice were isolated by using the differential adhesion method. Cells from the femurs were flushed out with fresh α-MEM (Gibco), passed through a 70-μm filter (BD Biosciences), and then centrifuged at 427 g for 10 min at 4 °C. The cells were resuspended in α-MEM supplemented with 15% (v/v) FBS and penicillin/streptomycin and cultured at 37 °C in a humidified atmosphere at 5% $CO_2$. After 2 days, non-adherent cells were washed with PBS and the remained cells were cultured under the same condition. BMSCs ($2 \times 10^5$) were seeded in 6-well plates and cultured in osteogenic differentiation basal medium (Cyagen, MUXMT-03021-175) for 14 days and 28 days. Then cells were fixed with 4% paraformaldehyde (PFA) and stained with an ALP detection kit (Beyotime, C3206) and 1% Alizarin red S (Solarbio, G1452).

### CFU-F and CFU-Ob assays

BMSCs ($1 \times 10^4$) were seeded in a 6-well plate and cultured in α-MEM supplemented with 15% (v/v) FCS or osteogenic differentiation basal medium (Cyagen, MUXMT-03021-175) for 14 days. Then, the cells were fixed with 4% PFA and stained with 1% Crystal Violet Ammonium Oxalate Solution (Solarbio, G1062).

### Calcein double labeling

8-week-old male mice were injected intraperitoneally with calcein (Sigma, 40 mg/kg) 3 and 10 day before skeleton collection. Femurs were fixed in 4% PFA and dehydrated in 30% sucrose at 4 °C overnight. Non-demineralized femurs were embedded and cut into 10 μm by using freezing microtome (RWD Minux FS800, China). Fluorescence signals were captured using a Pannoramic MIDI scanner (3 DHISTECH, Hungary). The mineral apposition rate (MAR) was analyzed by using Bioquant Osteo.

### Micro CT analysis and biomechanical test

Femurs from 8-week-old mice were fixed in 4% PFA for 48 h and then stored in 75% ethanol at 4 °C for Micro CT analysis according the previous study[62] and the femurs were subjected to a three-point bending test by using a universal testing device (CellScale Biomaterials Testing, Canada).

### Lentivirus infection

Lentiviruses carrying mouse HOIP full length or HOIP knockdown shRNA lentiviral vectors were obtained according to the previous study[2]. The viruses were concentrated with 5 × Lentivirus Concentration Reagent (GenStar, C103) and used to infect cells. Ninety-six hours later, infected cells were cultured in medium containing puromycin to select stable clones and verified by immunoblotting.

### Bacterial infection and cell adhesion assay

HeLa cells were infected with *Shigella flexner* (100: 1 MOI of *Shigella flexner* (OD600 = 0.8)) for 2 h and the adherent cells were stained with 0.1% crystal violet at the indicated time, followed by quantifications and photographs. For quantifications, the adherent cells were stained with 0.5% crystal violet for 30 min, and then the crystal violet was extracted with 1 ml 10% acetic acid for the quantification at 590 nm with Multiscan sky (Thermo Fisher Scientific).

### Bacterial Colony-forming units (CFU) assay

Cells were seeded in antibiotic-free complete medium. The following day, cells were infected with *S. flexneri* (MOI ~ 100), the bacteria were washed off with PBS 2 h post infection and placed in fresh DMEM containing 50 μg/ml gentamycin (to kill extracellular bacteria). After another 2 h of incubation, infected cells were washed three times with PBS, and placed in medium containing 10% FBS. After 12 h of infection, the attached and detached cells were collected and disrupted by the addition of sterile water with 1% Triton X-100, diluted, and plated onto agar plates and the number of internalized bacteria was determined by counting the colony-forming units after incubation at 37 °C.

### Immunohistochemical staining

Hydrated bone sections were stained for hematoxylin and eosin staining (Solarbio, G1120), TRAP staining (Solarbio, G1492) and

Safranin O/Fast green staining (Solarbio, G1371) according to the manufacturer's directions, and analyzed by the Bioquant Osteo software (Bioquant Image Analysis Corp, USA).

## Immunofluorescence

Immunofluorescence of bone sections were described in the previous studies[62]. Cell samples were fixed in 10% formalin, then stained with indicated antibodies, followed by Alexa Fluor 488/546/594 IgG (Invitrogen), and analyzed by fluorescence microscopy (NikonA1R) and NIS-Elements AR (4.40.00).

## RT-PCR and qPCR

BMSCs RNA was prepared using TRIzol reagent (Sigma, T9424) following the manufacturer's instructions. Total RNA was subjected to reverse transcription to synthesize cDNA using a ReverTra Ace® qPCR RT Master Mix (Toyobo, FSQ-201). qPCR was performed with QuantStudio 3 (Thermo Fisher Scientific) by using 2× RealStar Fast SYBR qPCR Mix (GenStar, A301). Genes specific primers were used to amplify cDNA. The sequences of primers were provided in Supplementary Data 8.

## Statistics and reproducibility

For in vitro study, investigators were not blinded to the sample identities during data collection since the readouts were quantitative and not prone to the subjective judgment of investigators. For in vivo study, mice experiments and statistical analysis were performed by independent researchers in a blinded manner. Statistical analyses were performed using GraphPad Prism 8.0.2. Statistical significance was determined using the two-tailed Student's $t$-test, two-way analysis of variance (ANOVA) followed by multiple comparisons and $p$ value were mentioned in the figure legends and were indicated in the figures respectively. The data were shown as the mean ± SEM, and additional details about numbers of samples were indicated in the corresponding figure legends. All the data were included at least three biological replicates.

## Reporting summary

Further information on research design is available in the Nature Portfolio Reporting Summary linked to this article.

## Data availability

Raw mass spectrometry data generated in this study have been deposited in ProteomeXchange Consortium via the iProX repository with the identifier PXD043322 and these data have been publicly released. All the data supporting the findings of this study are available within the article and its Supplementary Information. Source data have been deposited in the Figshare (https://figshare.com/) and the DOIs for Figshare are provided (https://doi.org/10.6084/m9.figshare.25303996, https://doi.org/10.6084/m9.figshare.25040426). Source data are also provided within this paper.

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

## Acknowledgements

This work was supported by the National Key Research and Development Project of China (2021YFA1300200, 2022YFC2302900), the National Natural Science Foundation of China (82192881, 32201023, 81825014, 82330069), Young Elite Scientists Sponsorship Program by CAST (YESS20220049) and the State Key Laboratory of Proteomics (SKLP-K202001). We would like to thank Prof. Guan Yang (Beijing Institute of Lifeomics) for the help in immunohistochemical staining.

## Author contributions

Lingqiang Zhang, Cui Hua Liu, and Yesheng Fu conceived the project. Lingqiang Zhang, Yesheng Fu and Cui Hua Liu designed the experiments. Yesheng Fu, Lei Li, Xin Zhang, Zhikang Deng, and Ying Wu performed most of the experiments. Jian Wang, Yuchen Liu, Yuping Xie and Zhiwei Tu carried out the MS and bioinformatics analysis. Shan He, Yadi Lyu, Shujie Wang, Wenzhe Chen and Yange Wei contributed the plasmid constructs, protein purifications and interaction analysis. Yesheng Fu, Lingqiang Zhang, Cui Hua Liu and Chun-Ping Cui analyzed the data. Yesheng Fu, Lingqiang Zhang and Cui Hua Liu wrote the manuscript.

## Competing interests

The authors declare no competing interests.
