## [Peer Review File · Nature Communications]

REVIEWER COMMENTS

Reviewer #1 (Remarks to the Author):

LUBAC, which is composed of HOIP, HOIL-1L, and SHARPIN subunits, is the only ubiquitin ligase (E3) to generate the Met1(M1)-linked linear ubiquitin chain, and HOIP includes the active site. In this manuscript, Fu and co-authors performed an exhaustive domain interactome analysis using portions of HOIP, such as the PUB, NZF, UBA, and RBR-LDD domains, from various mouse tissue lysates by GST-pulldown followed by mass spectrometry. Then, the authors identified an average of 90 interactors, including novel ones, for each domain per tissue. Furthermore, the authors investigated the involvement of linear ubiquitination of Integrin-linked kinase (ILK) and SMAD-specific E3 ubiquitin protein ligase1 (SMURF1) on cell defense against *Shigella* infection and osteogenesis, respectively. This study comprehensively analyzed proteins that may bind to HOIP in the steady state, identified potential new substrates, and clarified some of their physiological functions, which is very important. In this manuscript, most of the analyses of binding and ubiquitination in overexpression systems lack evidence at the endogenous level. Analysis of linear ubiquitination has unclear points and needs to be improved. In addition, since this study is limited to steady-state conditions, further investigations of the alteration of interactome in pathological models accompanied by LUBAC activation and their intracellular verification are desired.

Major comments:

1. In Figure 1g, h: It is interesting to note that the binding of STAT1 and β -catenin is tissue-specific and that the interaction domains of HOIP between these proteins vary depending on the tissue. This may be due to direct or indirect interaction through another protein. If not, the post-translational modifications of these proteins may differ between tissues. Is it possible to clarify the STAT1 and β -catenin interacting factors and binding domain in HOIP in each tissue that cause such differences? Also, does the endogenous STAT1 or β -catenin binding to endogenous LUBAC differ in each tissue lysate or tissue-derived cell line?
2. Tissue-specific HOIP PPIs are very interesting. There isn't much difference in inputs, but is it possible that differences in protein expression levels in different tissues affect HOIP binding? It would be wonderful if the authors could analyze how the HOIP interactome changes under LUBAC activation, such as in LPS-induced sepsis models, even in specific tissues.
3. The authors describe HOIP binding and linear ubiquitination equivalently in many places in the text, showing the results in Figures 2 and 3. Since the intracellular content of linear ubiquitin chains is extremely low in the unstimulated state, proteins that transiently bind upon stimulation are likely substrates for LUBAC. Therefore, whether the steady-state binding proteins are substrates for linear ubiquitination is still being determined. In Fig. 3c–e and Figures using similar analysis methods, the authors analyze whether HOIP-binding proteins can be substrates for linear ubiquitination in an overexpression system. However, clarifying covalent ubiquitination requires heat denaturation in 1% SDS, dilution in 1% Triton X-100, immunoprecipitation, and immunoblotting. The authors must also demonstrate negative controls that are not ubiquitinated in the presence of HOIP-CS.
4. LUBAC and OTULIN are known to be involved in autophagy (xenophagy) regulation of bacteria that have invaded cells. In addition, ILK is known to be involved in autophagy in *Helicobacter pylori*. Are ILK activity and linear ubiquitination in *Shigella*-infected cells in Figure 4 possibly involved in autophagy?

Furthermore, clarifying the effects of Shigella infection in cells that expressed a non-linear ubiquitinated ILK mutant (2KR) is essential. Do OspE, ILK, and linear ubiquitin co-localize on the basal side in Shigella-infected cells?

5. Recently, it is considered necessary that several E3s cooperate to generate complex ubiquitin chains, such as branched chains and mixed chains. What happens to SMAD1/5 ubiquitination when co-expressing not only HOIP but also all LUBAC components and SMURF1 in Fig. 5e? Consider the possibility that complex ubiquitin chains are added to SMAD1/5 using SMURF1-4KR and HOIP-CS as well. Please clarify the effect on TGF- β /BMP/SMAD signaling using HOIP-KO or -knockdown cells.

6. Show expression levels of HOIL-1L and SHARPIN in Supplementary Figures 6d or 6i. Can immunohistochemical staining using anti-HOIP, anti-linear ubiquitin, and SMAD signaling factor antibodies show where these crosstalks are essential in osteogenesis?

Minor comments:

1. Line 121 and Fig. 1f: Please clarify whether it indicates PI3K or Wnt.
2. In Supplementary Fig. 4j, α -Tubulin should be input.
3. Line 319: Binding between SMURF1 and HOIP has also been detected in the brain.
4. Line 331: A decrease in MEKK2 protein levels with HOIP knockdown has not been shown.
5. Line 336: Doesn't linear ubiquitination of SMURF1 increase SMAD1/5?

Reviewer #2 (Remarks to the Author):

In this manuscript Yesheng Fu and colleagues identify interacting partners of the E3 ligase HOIP, the catalytic member of the LUBAC complex, an enzyme with pleiotropic functions.

To do this, the authors use affinity-purification mass spectrometry: they purify four domains of the HOIP protein (all fused to GST) and use these as bait to identify interacting partners in 9 different tissues (as distinct as brain, spleen, and bone, for example). This yields approximately 3500 unique protein-protein interactions (PPIs), some of which are shared between tissues and the domains. The authors confirmed some of their PPIs using in situ immunoprecipitation. These included interaction of HOIP with proteins involved in synapse movement, myocardial function, nucleosome remodelling, glycolysis. The authors confirm the linear ubiquitination of these new interacting partners, such as ILK (whose linear ubiquitination might be important against Shigella flexneri infection) and SMURF1 (whose linear ubiquitination is important for normal bone growth).

Interestingly, the authors observe that most interacting partners of HOIP were defined primarily by the tissue, not by the domain that was used (line 219-220: "Approximately 50% of the detected HOIP PPIs were found across the four domains"). This suggests that the different domains are redundant for the interacting partner "selection" or that the tag might be interfering with the PPIs. It would be ideal that the authors confirm which is true using a different tag (please see Major Points section).

One downside is that the paper is quite dense. This is understandable given the amount of information generated and the limited space available in a manuscript. Yet, it would benefit from a bit more careful writing. For example, the reader can infer from the text that the authors identify approximately 3500 total PPIs (line 108-109 "uncovered an average of approximately 90 interactors for each bait in a tissue"; and lines 143-144 "we observed 109 (3.1%) tissue-shared PPIs and 1817 (52.9%) potential tissue-specific

PPIs”) but I think this information should be explicit in the writing. One hypothesis would be to remove one of the models (infection or bone growth), which give the authors more space on this paper, and publish the remaining results in another publication.

Nonetheless, the results reported in this paper are strong and the interactions reported may inspire multiple future projects. The following points should be addressed to improve the manuscript:

Major points:

- The title of the paper is not reflecting the main findings properly. The main finding of the paper is the database of potential interactors that were discovered for nine different tissues, not the infection or the osteogenesis. Consider changing the title.
- The authors should state the total number PPIs identified as well as of unique PPIs identified.
- In the graphs in Figure 1c and 1d why are the bars of the same pair of different sizes in Figure 1c and 1d? What is the 100% referring to? I would prefer to have the absolute numbers of PPI discovered for each organ and bait pair, even if some of these are shared between organs and domains.
- The authors say that most of the PPIs identified are shared between the different domains, which I find very surprising. Give that all the domains were tagged to GST, did the authors use a different tag/reporter protein in their affinity purification-MS pipeline to exclude any effect of the GST tag? Alternatively, the authors could confirm this effect using using myc- or FLAG-tagged domains of HOIP and not just the full-length protein in cellulose.
- What is the level of ILK in the input of cells treated with shRNA against HOIP? And in infected cells throughout infection? From the immunofluorescence image, it seems that ILK levels are lower after shRNA treatment in basal conditions but not after infection; and in control infected cells seem to lose ILK. If this is true, shouldn't the shRNA-treated cells have worse attachment potential? Does the infection modulate ILK expression?
- The phosphorylation of FAK (Figure 4d) still decreases in the infected and shRNA-treated cells. Could this happen because ILK is being ubiquitinated by other E3 ligases and with other Ubiquitin-linkages?
- If HOIP::ILK interaction serves as a cell-autonomous mechanism against infection, the authors should measure bacterial load in the attached cells (and ideally in detached cells) after 12h of *S. flexneri* infection.
- In Figure 5d, in the whole cell lysate blots, why is there a band for HA when no HA-HOIL1 was transfected?

Minor points:

- The order of the panels in most Figures is confusing and difficult to follow. For example, in Figure 2, panels a and b are in the top left, c is in the middle left, d is in the top right, e is bottom and f is middle right. The Figures would be easier to follow if the panel would be more coherently organized. A good template is Figure 6, panels go from left to right first, then top to bottom.
- In most tissues, the PPIs from the different domains correlate quite nicely. This is not the case for the spleen and the heart. This is also true for the NZF domain when compared to the other domains (Liver, Brain, and Colorectum). Can the authors elaborate on this in the discussion?
- The label “Colorectum” is missing from the bottom axis of the plot in Figure 2a.
- In terms of the PPIs identified. Lung, Colorectal and Stomach are closely related. Can the authors comment on this in the discussion?
- Figure 3a refers to all PPIs that were found, or only the 50% that are common across the 4 domains?

- In Figure 4h, the authors should report the actual confluency along time, not the normalization.
- There is a typo in Figure 7, bottom left rectangle, it should read “M1 polyUb” instead of “M1 ployUb”.
- There are yellow markings throughout the Materials and Methods section.

Reviewer #3 (Remarks to the Author):

In this manuscript, Fu et al. sought to analyze the physiological function of the linear ubiquitin E3 ligase LUBAC. For this purpose, the authors examined the tissue-specific interaction landscape of the catalytic LUBAC component HOIP. Using key functional domains of HOIP purified from bacteria as baits, the authors performed pull-downs in lysates from nine different murine tissues and analyzed bound proteins by quantitative mass spectrometry. Using stringent filtering, the authors identified common and tissue-specific candidate interacting proteins for different functional domains of HOIP. The resulting complex data sets (i) a number of known interaction partners whose tissue specificity the authors validated biochemically as well as (ii) a range of proteins whose gene annotations reflect functions assigned to certain tissues. Focusing on identifying potential HOIP substrates, the authors employed immunoprecipitations (IP) and pull-downs to validate the binding of an impressive number of candidates to full-length HOIP and its substrate binding domain NZF. Using overexpression approaches, the authors verified linear ubiquitination of selected NZF-binding proteins as potential substrate candidates. From the pool of candidate HOIP binders and substrates, the authors decided to characterize the role of LUBAC in regulating the kinase ILK and the E3 ligase SMURF1. While the authors showed that linear ubiquitination of ILK contributes to the defense against bacterial infection, M1-linked ubiquitin modification of SMURF1 controls the activity of SMURF1. Lastly, the authors generated and characterized an osteoblast-specific HOIP deletion mouse model. Overall, the work of Fu and colleagues provides a rich protein-protein interaction (PPI) data set for HOIP and demonstrates that this resource can be exploited to identify LUBAC targets. However, several concerns remain.

1) What did the authors use as negative control for their proteomics analysis? From the method section it seems that no dedicated control such as GST alone was used. That raises concerns to what extent the data set contains high numbers of false positive candidates and background binding proteins. Therefore, the authors should generate a reference data set and repeat the pull-downs with all different tissues using GST alone or GST fused to an unrelated protein.

2) Along similar lines, it is not clear what the authors used as negative control in the IP experiments shown in Figure 2d and 3b-E, 5A? This should be clearly stated in the figure legend. If it is MOCK transfection, then the authors should use an unrelated myc/Flag-tagged protein for their IPs instead.

3) The ubiquitination assays in Figure 3C-E, 4B-D and 4I as well as 5C-E and 5H should be repeated under denaturing conditions otherwise the authors cannot conclude that the respective substrate candidates are covalently modified by linear ubiquitin.

4) For ILK and SMURF1 the authors need to provide evidence that these substrate candidates are indeed linearly ubiquitinated by LUBAC at endogenous levels. While Figure 4d is meant to show altered

ubiquitination levels of ILK in a LUBAC dependent manner. However, this experiment needs to be repeated under denaturing conditions and using a M1-Ub specific antibody.

5) Figure 6 is completely disconnected from the rest of the manuscript. While the observed phenotypes are interesting, there is no link that these defects are dependent on linear ubiquitination of SMURF1.

6) Throughout the manuscript the authors need to tone down any conclusions of linear ubiquitination substrates (except in the case of ILK) since the authors mainly show PPI data and no ubiquitination assays under stringent conditions.

Response letter

Point-by-point responses to the referees' comments

RE: Fu et al., NCOMMS-23-35622, "HOIP Interactome Profiling Reveals Critical Roles of Linear Ubiquitination in Bacterial Infection and Osteogenesis".

Reviewer #1 (Remarks to the Author):

LUBAC, which is composed of HOIP, HOIL-1L, and SHARPIN subunits, is the only ubiquitin ligase (E3) to generate the Met1(M1)-linked linear ubiquitin chain, and HOIP includes the active site. In this manuscript, Fu and co-authors performed an exhaustive domain interactome analysis using portions of HOIP, such as the PUB, NZF, UBA, and RBR-LDD domains, from various mouse tissue lysates by GST-pulldown followed by mass spectrometry. Then, the authors identified an average of 90 interactors, including novel ones, for each domain per tissue. Furthermore, the authors investigated the involvement of linear ubiquitination of Integrin-linked kinase (ILK) and SMAD-specific E3 ubiquitin protein ligase1 (SMURF1) on cell defense against Shigella infection and osteogenesis, respectively. This study comprehensively analyzed proteins that may bind to HOIP in the steady state, identified potential new substrates, and clarified some of their physiological functions, which is very important. In this manuscript, most of the analyses of binding and ubiquitination in overexpression systems lack evidence at the endogenous level. Analysis of linear ubiquitination has unclear points and needs to be improved. In addition, since this study is limited to steady-state conditions, further investigations of the alteration of interactome in pathological models accompanied by LUBAC activation and their intracellular verification are desired.

Response: We thank the reviewer for the insightful comments on our manuscript. We also thank the reviewer for providing valuable suggestions to help us improve the manuscript.

Major comments:

1. In Figure 1g, h: It is interesting to note that the binding of STAT1 and β -catenin is tissue-specific and that the interaction domains of HOIP between these proteins vary depending on the tissue. This may be due to direct or indirect interaction through another protein. If not, the post-translational modifications of these proteins may differ

between tissues. Is it possible to clarify the STAT1 and β -catenin interacting factors and binding domain in HOIP in each tissue that cause such differences? Also, does the endogenous STAT1 or β -catenin binding to endogenous LUBAC differ in each tissue lysate or tissue-derived cell line?

2. Tissue-specific HOIP PPIs are very interesting. There isn't much difference in inputs, but is it possible that differences in protein expression levels in different tissues affect HOIP binding? It would be wonderful if the authors could analyze how the HOIP interactome changes under LUBAC activation, such as in LPS-induced sepsis models, even in specific tissues.

Response to major comments 1, 2:

We appreciate these valuable suggestions by the reviewer.

1) As shown in revised Supplementary Fig. 1f, endogenous STAT1 bound to endogenous HOIP in liver, lung and colorectum instead of other tissues. Moreover, endogenous β -catenin specifically bound to endogenous LUBAC in brain, liver, lung, stomach. These endogenous interactions of STAT1 and β -catenin were almost the same as the binding patterns of GST-HOIP truncations in different tissues as shown in previous Figs. 1h, i.

2) To figure out the potential mechanism for tissue-specific HOIP PPIs, the protein expression levels of these interacting proteins were detected according to the reviewer's suggestion. As shown in revised Supplementary Fig. 1g, the tissue-specific binding pattern of HOIP and β -catenin in brain, liver, lung and colorectum was probable due to the higher expression levels of β -catenin. Moreover, another HOIP-interacting protein FXR1, a previously reported β -catenin interactor (Li et al., 2017, *Nat Neurosci.*, 20(8):1150-1161), was highly expressed in the stomach, which might contribute to the stomach-specific binding of HOIP and β -catenin (revised Supplementary Figs. 1g, h). All of these data suggest that protein expression level in different tissues is a potential factor that affects HOIP binding.

3) According to the reviewer's suggestions, we conducted the HOIP lung-specific interactome in LPS-induced sepsis models. As shown in revised Supplementary Figs. 5a-c, LPS-induced sepsis model (i.p. injection, 12 mg/kg body weight in saline) was successfully conducted with decreased body mass, increased expression level of inflammatory factors and lung injuries. Further analysis of HOIP interactome in LPS-induced sepsis models (revised Supplementary Fig. 5d) showed that LPS treatment upregulated some significant GO terms such as "cellular response to interleukin-7",

“defense response to bacterium”, “positive regulation of protein localization to telomere” and “proteasome-mediated ubiquitin-dependent protein catabolic process”. The last GO term “proteasome functions” has been validated in a previous report (Tao et al., *Sci. Adv.* 7, eabi6794 (2021)). Other upregulated or downregulated GO terms warrant further study in the future.

3. The authors describe HOIP binding and linear ubiquitination equivalently in many places in the text, showing the results in Figures 2 and 3. Since the intracellular content of linear ubiquitin chains is extremely low in the unstimulated state, proteins that transiently bind upon stimulation are likely substrates for LUBAC. Therefore, whether the steady-state binding proteins are substrates for linear ubiquitination is still being determined. In Fig. 3c–e and Figures using similar analysis methods, the authors analyze whether HOIP-binding proteins can be substrates for linear ubiquitination in an overexpression system. However, clarifying covalent ubiquitination requires heat denaturation in 1% SDS, dilution in 1% Triton X-100, immunoprecipitation, and immunoblotting. The authors must also demonstrate negative controls that are not ubiquitinated in the presence of HOIP-CS.

Response: We fully agree with the reviewer on this point and thank for this good suggestion. According to the reviewer’s suggestions, we repeated the linear ubiquitination experiment with strict heat denaturation methods by using HOIP-CS as a negative control and provided better data in revised Supplementary Figs. 3e, f.

4. LUBAC and OTULIN are known to be involved in autophagy (xenophagy) regulation of bacteria that have invaded cells. In addition, ILK is known to be involved in autophagy in *Helicobacter pylori*. Are ILK activity and linear ubiquitination in *Shigella*-infected cells in Figure 4 possibly involved in autophagy? Furthermore, clarifying the effects of *Shigella* infection in cells that expressed a non-linear ubiquitinated ILK mutant (2KR) is essential. Do OspE, ILK, and linear ubiquitin co-localize on the basal side in *Shigella*-infected cells?

Response: We thank the reviewer for bringing up these good suggestions.

1) To figure out the roles of ILK activity and linear ubiquitination in *Shigella*-infected cells possibly involved in autophagy, ILK inhibitor ILK-IN-3 (Selleck, E0790, 2 μ M) and HOIP inhibitor HOIPin-8 (20 μ M, Ken Katsuya et al, 2018) were used to treat *Shigella*-infected cells. Compared with untreated *Shigella*-infected cells, ILK inhibitor

enhanced the autophagy of cells while HOIP inhibitor failed (revised Supplementary Fig. 7j). This result suggested that ILK activity was involved in autophagy regulation of *Shigella flexneri* infection while LUBAC was not.

2) According to the reviewer's suggestions, the effects of *Shigella* infection in cells that expressed a non-linear ubiquitinated ILK mutant (2KR) were tested. As shown in revised Fig. 4j and Supplementary Fig. 7n, the adhesion of ILK mutant cells was higher than that in wildtype cells. Moreover, OspE, ILK, and linear ubiquitin co-localized on the basal side in *Shigella*-infected cells (revised Supplementary Fig. 7o). All of these data was consistent with our hypothesis that linear ubiquitination promoted the detachment of *Shigella*-infected cells.

5. Recently, it is considered necessary that several E3s cooperate to generate complex ubiquitin chains, such as branched chains and mixed chains. What happens to SMAD1/5 ubiquitination when co-expressing not only HOIP but also all LUBAC components and SMURF1 in Fig. 5e? Consider the possibility that complex ubiquitin chains are added to SMAD1/5 using SMURF1-4KR and HOIP-CS as well. Please clarify the effect on TGF- β /BMP/SMAD signaling using HOIP-KO or -knockdown cells.

Response: We thank the reviewer for this good suggestion.

1) As shown in revised Supplementary Fig. 8g, linear ubiquitination of SMURF1 instead of SMAD1/5 was detected in our experiment condition. Moreover, linear ubiquitination conjugated by LUBAC indeed decreased the ubiquitination level of SMAD5 while SMURF1-4KR and HOIP inhibitor failed (revised Fig. 5l).

2) According to the reviewer's suggestion, TGF- β /BMP/SMAD signaling in HOIP-CKO mice derived BMSCs was detected and the result (revised Fig. 7f) showed that p-SMAD1/5 and SMAD1/5 (components of BMP signaling) levels decreased in HOIP-CKO BMSCs while p-SMAD2/3 and SMAD2/3 (components of TGF- β signaling) levels remained unchanged.

6. Show expression levels of HOIL-1L and SHARPIN in Supplementary Figures 6d or 6i. Can immunohistochemical staining using anti-HOIP, anti-linear ubiquitin, and SMAD signaling factor antibodies show where these crosstalk are essential in osteogenesis?

Response: This is a good suggestion. We measured the expression of HOIL-1L and

SHARPIN in revised Fig. 7f and Supplementary Fig. 9d. Moreover, immunohistochemical staining of linear ubiquitination, RUNX2 (the osteoblast cell marker) and SMURF1 (or SMAD1/5) was conducted. As showed in revised Figs. 7d, e, the signal of RUNX2 positive osteoblasts were decreased in femurs from HOIP CKO mice as expected and the colocalization of linear ubiquitination and SMURF1 (or SMAD1/5) in osteoblasts were also decreased in these HOIP CKO femurs.

Minor comments:

1. Line 121 and Fig. 1f: Please clarify whether it indicates PI3K or Wnt.
2. In Supplementary Fig. 4j, α -Tubulin should be input.
3. Line 319: Binding between SMURF1 and HOIP has also been detected in the brain.
5. Line 336: Doesn't linear ubiquitination of SMURF1 increase SMAD1/5?

Response to minor comments 1, 2, 3, 5: We thank you for pointing out these issues. We have corrected them in the revised manuscript and Figures.

4. Line 331: A decrease in MEKK2 protein levels with HOIP knockdown has not been shown.

Response: We thank the reviewer for raising this concern. We repeated this assay and provided better data in revised Fig. 5g.

Reviewer #2 (Remarks to the Author):

In this manuscript Yesheng Fu and colleagues identify interacting partners of the E3 ligase HOIP, the catalytic member of the LUBAC complex, an enzyme with pleiotropic functions.

To do this, the authors use affinity-purification mass spectrometry: they purify four domains of the HOIP protein (all fused to GST) and use these as bait to identify interacting partners in 9 different tissues (as distinct as brain, spleen, and bone, for example). This yields approximately 3500 unique protein-protein interactions (PPIs), some of which are shared between tissues and the domains. The authors confirmed some of their PPIs using in situ immunoprecipitation. These included interaction of HOIP with proteins involved in synapse movement, myocardial function, nucleosome remodeling, glycolysis. The authors confirm the linear ubiquitination of these new interacting partners, such as ILK (whose linear ubiquitination might be important

against *Shigella flexneri* infection) and SMURF1 (whose linear ubiquitination is important for normal bone growth).

Interestingly, the authors observe that most interacting partners of HOIP were defined primarily by the tissue, not by the domain that was used (line 219-220: “Approximately 50% of the detected HOIP PPIs were found across the four domains”). This suggests that the different domains are redundant for the interacting partner “selection” or that the tag might be interfering with the PPIs. It would be ideal that the authors confirm which is true using a different tag (please see Major Points section).

One downside is that the paper is quite dense. This is understandable given the amount of information generated and the limited space available in a manuscript. Yet, it would benefit from a bit more careful writing. For example, the reader can infer from the text that the authors identify approximately 3500 total PPIs (line 108-109 “uncovered an average of approximately 90 interactors for each bait in a tissue”; and lines 143-144 “we observed 109 (3.1%) tissue-shared PPIs and 1817 (52.9%) potential tissue-specific PPIs”) but I think this information should be explicit in the writing. One hypothesis would be to remove one of the models (infection or bone growth), which give the authors more space on this paper, and publish the remaining results in another publication.

Nonetheless, the results reported in this paper are strong and the interactions reported may inspire multiple future projects. The following points should be addressed to improve the manuscript:

Response: We thank the reviewer for the kind comments and constructive suggestions on our manuscript and for recognizing the novelty and significance of this study.

Major points:

1. The title of the paper is not reflecting the main findings properly. The main finding of the paper is the database of potential interactors that were discovered for nine different tissues, not the infection or the osteogenesis. Consider changing the title.

Response: We thank the reviewer for raising this issue. We have changed the title as “Systematic HOIP Interactome Profiling Reveals Critical Roles of Linear Ubiquitination in Tissue Homeostasis” in the revised manuscript.

2. The authors should state the total number PPIs identified as well as of unique PPIs identified.

Response: We thank the reviewer for raising this issue. In total, we identified 17078 high-confidence interactions with non-redundant 3430 proteins across domains and tissues. The unique PPIs in each organ and domain have been provided in the revised Figs. 1c, d.

3. In the graphs in Figure 1c and 1d why are the bars of the same pair of different sizes in Figure 1c and 1d? What is the 100% referring to? I would prefer to have the absolute numbers of PPI discovered for each organ and bait pair, even if some of these are shared between organs and domains.

Response: It is a really important point. As shown in revised Figs. 1c, d, the absolute numbers of HOIP PPIs discovered for each organ and bait pair have been provided.

4. The authors say that most of the PPIs identified are shared between the different domains, which I find very surprising. Give that all the domains were tagged to GST, did the authors use a different tag/reporter protein in their affinity purification-MS pipeline to exclude any effect of the GST tag? Alternatively, the authors could confirm this effect using myc- or FLAG-tagged domains of HOIP and not just the full-length protein in cellular.

Response: We thank the reviewer very much for pointing out this important issue. According to previous studies (Hein et al., 2015, *Cell* 163, 712-723; Huttlin et al., 2021, *Cell* 184(11):3022-3040.e28), the negative control always used the corresponding tagged vectors. Actually, in our assays, GST alone were purified from bacteria BL21 and used as a negative control for the subsequent pull-down and mass spectrometry analysis in tissues (shown in previous Fig. 7). Moreover, protein enrichment in the pulldowns of each HOIP-truncated protein versus GST alone were calculated and determined with the fold-over-control (FOC) data and p value in our analysis. All above description have been provided in the revised manuscript, methods and Figures.

Moreover, according to the reviewer's suggestion, HEK293T transfected with Myc-tagged vector or Myc-tagged HOIP-NZF plasmid was used to test the identified GO term "RNA splicing". The result (revised Supplementary Fig. 3d) showed the similar interaction pattern to HOIP full-length protein in cellular. The fact that most of the PPIs identified were shared between the different domains might be due to the indirect interaction identified by our pulldown-MS methods.

5. What is the level of ILK in the input of cells treated with shRNA against HOIP? And in infected cells throughout infection? From the immunofluorescence image, it seems that ILK levels are lower after shRNA treatment in basal conditions but not after infection; and in control infected cells seem to lose ILK. If this is true, shouldn't the shRNA-treated cells have worse attachment potential? Does the infection modulate ILK expression?

Response: It is a really important point. We repeated the immunofluorescence assay and counted the ILK levels in focal adhesion areas. As shown in revised Figs. 4d, e, the ILK levels in focal adhesion areas were higher in *shHOIP* groups when compared with control groups, and similar results were observed in *Shigella flexneri* infected groups. However, the total ILK protein levels in *shHOIP* groups remained unchanged (revised Fig. 4c), suggesting that the infection did not affect the ILK expression, but it modulated ILK localization in focal adhesion areas.

6. The phosphorylation of FAK (Figure 4d) still decreases in the infected and shRNA-treated cells. Could this happen because ILK is being ubiquitinated by other E3 ligases and with other Ubiquitin-linkages?

Response: We thank the reviewer for raising this concern. We have repeated this experiment three times and provided quantitative data here. As shown here and in revised Fig. 4c, the phosphorylation of FAK remained unchanged in the infected *shHOIP* group.

7. If HOIP::ILK interaction serves as a cell-autonomous mechanism against infection, the authors should measure bacterial load in the attached cells (and ideally in detached cells) after 12h of *S. flexneri* infection.

Response: We thank the reviewer very much for the kind comments and the insightful concerns. As suggested, the bacterial load in the attached cells and detached cells after 12h of *Shigella flexneri* infection were detected. As shown in revised Figs. 4h, i, the bacterial load in the attached cells increased in the *shHOIP* group and bacterial load in the detached cells decreased in the *shHOIP* group. Similar results were observed in ILK-2KR (mutation of linear ubiquitinated sites) groups (revised Figs. 4k, l).

8. In Figure 5d, in the whole cell lysate blots, why is there a band for HA when no HA-HOIL1 was transfected?

Response: We thank the reviewer for pointing out these errors. We repeated this assay with the more strict condition and provided correctly marked data in revised Fig. 5e.

Minor points:

1. The order of the panels in most Figures is confusing and difficult to follow. For example, in Figure 2, panels a and b are in the top left, c is in the middle left, d is in the top right, e is bottom and f is middle right. The Figures would be easier to follow if the panel would be more coherently organized. A good template is Figure 6, panels go from left to right first, then top to bottom.

Response: We fully agree with this advice. We have rearranged our data in revised figures.

2. In most tissues, the PPIs from the different domains correlate quite nicely. This is not the case for the spleen and the heart. This is also true for the NZF domain when compared to the other domains (Liver, Brain, and Colorectum). Can the authors elaborate on this in the discussion?

Response: We thank the reviewer very much for pointing out these important issues and we analyzed GO terms specifically enriched in these tissues and domains. As shown in revised Supplementary Data 5d, PPIs from the different domains in spleen showed very good specificity: the term “cytokine production involved in inflammatory response” was enriched in the PUB domain, the term “tRNA processing” and “negative regulation of miRNA-mediated gene silencing” (more related to transcription) were enriched in

NZF domain and the term “regulation of immunoglobulin production” was enriched in the UBA domain, as well as the term “detection of chemical stimulus” was enriched in the RBR domain. Specific GO terms were also observed in NZF domain of brain, liver and colorectum. All of these revealed an unexpected pattern of HOIP domains across tissues, which was probably caused by the post-translational modifications in HOIP domains or the tissue-specific expression of HOIP interactors. For example, the tissue-specific binding pattern of HOIP and β -catenin in brain, liver, lung and colorectum was probable due to the higher expression levels of β -catenin in these tissues. Moreover, another HOIP-interacting protein FXR1, a previous reported β -catenin interactor (Li et al., 2017, *Nat Neurosci.*, 20(8):1150-1161), was highly expressed in the stomach, which might contribute to the stomach-specific binding of HOIP and β -catenin (revised Supplementary Figs. 1h, i).

3. In terms of the PPIs identified. Lung, Colorectal and Stomach are closely related. Can the authors comment on this in the discussion?

Response: We thank the reviewer for raising this outstanding concern and we have provided the details in the revised manuscript. The close relation among lung, colorectum and stomach was partially due to the exposure of these tissues to the external environment, which caused the enrichment of similar immune-related processes, such as “type I interferon production (revised Supplementary Fig. 2a).

4. Figure 3a refers to all PPIs that were found, or only the 50% that are common across the 4 domains?

Response: We thank the reviewer for pointing out these issues. Figure 3a showed selected networks of similar biological processes (terms significantly enriched in all domains) enriched in all HOIP PPIs. Proteins involved in these processes associated with each domain have been shown in the revised Supplementary Fig. 3c (previous Supplementary Fig. 3c).

5. In Figure 4h, the authors should report the actual confluency along time, not the normalization.

Response: We thank the reviewer for raising this concern. Actually, we used OD590 to report the number of adhesion cells according to previous studies (Dirac et al., 2003, *Journal of Biological Chemistry*, 278(14): 11731-11734; García et al., 2002,

Oncogene 21, 8379–8387). In details, the adherent cells after *S. flexneri* infection were stained with 0.5% crystal violet for 30 min, and then the crystal violet was extracted with 1 ml 10% acetic acid for quantification at 590 nm with Multiscan sky (Thermo Fisher Scientific). All of these details have been provided in revised Figs. 4f and 4j.

6. The label “Colorectum” is missing from the bottom axis of the plot in Figure 2a.

7. There is a typo in Figure 7, bottom left rectangle, it should read “M1 polyUb” instead of “M1 ployUb”.

8. There are yellow markings throughout the Materials and Methods section.

Response to minor points 6, 7, 8: We thank the reviewer for pointing out these issues. We have corrected them in the revised manuscript and figures.

Reviewer #3 (Remarks to the Author):

In this manuscript, Fu et al. sought to analyze the physiological function of the linear ubiquitin E3 ligase LUBAC. For this purpose, the authors examined the tissue-specific interaction landscape of the catalytic LUBAC component HOIP. Using key functional domains of HOIP purified from bacteria as baits the authors performed pulldowns in lysates from nine different murine tissues and analyzed bound proteins by quantitative mass spectrometry. Using stringent filtering the authors identified common and tissue-specific candidate interacting proteins for different functional domains of HOIP. The resulting complex data hosts (i) a number of known interaction partners whose tissue specificity the authors validated biochemically as well as (ii) a range of proteins whose gene annotations reflect functions assigned to certain tissues. Focusing on identifying potential HOIP substrates, the authors employed immunoprecipitations (IP) and pulldowns to validate the binding of an impressive number of candidates to full-length HOIP and its substrate binding domain NZF. Using overexpression approaches, the authors verified linear ubiquitination of selected NZF-binding proteins as potential substrate candidates. From the pool of candidate HOIP binders and substrates, the authors decided to characterize the role of LUBAC in regulating the kinase ILK and the E3 ligase SMURF1. While the authors showed that linear ubiquitination of ILK contributes to the defense against bacterial infection, M1-linked ubiquitin modification of SMURF1 controls the activity of SMURF1. Lastly, the authors generated and characterized an osteoblast-specific Hoip deletion mouse model. Overall, the work of Fu

and colleagues provides a rich protein-protein interaction (PPI) data set for HOIP and demonstrate that this resource can be exploited to identify LUBAC targets. However, several concerns remain.

Response: We thank the reviewer for kindly commenting on the thoroughness of our manuscript and proving constructive suggestions to help improve the manuscript.

1) What did the authors used as negative control for their proteomics analysis? From the method section it seems that no dedicated control such as GST alone was used. That raises concerns to what extend the data set contains high numbers of false positive candidates and background binding proteins. Therefore, the authors should generate a reference data set and repeat the pulldowns with all different tissues using GST alone or GST fused to an unrelated protein.

Response: We thank the reviewer very much for pointing out this important issue. We apologized for not providing sufficient description of the criteria of HOIP PPIs before. Actually, GST alone and GST-fused HOIP-truncated proteins were both purified from bacteria BL21 and used for subsequent pull-down and mass spectrometry analysis in mouse tissues (shown in previous Fig. 7). Besides, according to previous studies (Hein et al., 2015, *Cell* 163, 712–723; Hubel et al., 2019, *Nat Immunol.*, 20(4):493-502), protein enrichment in the pulldowns of each HOIP-truncated protein versus GST alone were calculated and determined with the fold-over-control (FOC) data and p value. HOIP PPIs were defined with cutoffs of an $FOC \geq 2$ and a $P_{FOC} \leq 0.05$. All the above description has been provided in the revised manuscript, methods and figures.

2) Along similar lines, it is not clear what the authors used as negative control in the IP experiments shown in Figure 2d and 3b-E, 5A? This should be clearly stated in the figure legend. If it is MOCK transfection, then the authors should use an unrelated myc/Flag-tagged proteins for their IPs instead.

Response: We thank the reviewer very much for pointing out these important issues. We apologized for not providing sufficient description of figure legends. As shown in previous Figs.2d, 3b and 5a, Flag-tagged peptide or GST alone protein were used for IPs. And in previous Fig.3b, cells were transfected with Myc-tagged vector as a negative control. All the above description has been provided in the revised manuscript and figure legends.

3) The ubiquitination assays in Figure 3C-E, 4B-D and 4I as well as 5C-E and 5H should be repeated under denaturing conditions otherwise the authors cannot conclude that the respective substrate candidates are covalently modified by linear ubiquitin.

Response: We fully agree with the reviewer on this point. According to the suggestions of two reviewers, we repeated all the ubiquitination assays with a strict condition (the cell samples were heat denaturation in 1% SDS, dilution in TNE buffer (containing 1% NP-40, 1% Triton X-100 and 0.5% sodium deoxycholate), immunoprecipitation, and immunoblotting) and have provided better data in revised Figs. 4c, 4m, 5d-f, 5k, 5l and Supplementary Figs. 3e, 3f, 7d and 8g. All of these data suggested that these respective substrate candidates were covalently modified by linear ubiquitin.

4) For ILK and SMURF1 the authors need to provide evidence that these substrate candidates are indeed linearly ubiquitinated by LUBAC at endogenous levels. While Figure 4d is meant to show altered ubiquitination levels of ILK in a LUBAC dependent manner. However, this experiment needs to be repeated under denaturing conditions and using a M1-Ub specific antibody.

Response: We thank the reviewer very much for pointing out these important issues. As shown in revised Figs. 4c, 5f and 7g, ILK and SMURF1 were indeed linearly ubiquitinated by LUBAC by using denaturing conditions.

5) Figure 6 is completely disconnected from the rest of the manuscript. While the observed phenotypes are interesting, there is no link that these defects are dependent on linear ubiquitination of SMURF1.

Response: We thank the reviewer for raising this outstanding concern. To address this issue, immunohistochemical staining of linear ubiquitination, RUNX2 (the osteoblast cell marker) and SMURF1 was conducted. As showed in revised Figs. 7d, e, the signal of RUNX2 positive osteoblasts were decreased in femurs from HOIP CKO mice as expected and the colocalization between linear ubiquitination and SMURF1 in osteoblasts were also decreased in these femurs of HOIP CKO mice. Linear ubiquitination of SMURF1 attenuated its E3 activity and thus the E3 activity of SMURF1 towards SMAD1/5 was enhanced in HOIP knockdown cells (revised Figs 5g-l). Therefore, BMSCs derived from femurs of HOIP CKO mice were treated with A17 (SMURF1 inhibitor, 20 μ M) for 28 days to test whether the E3 ligase activity of SMURF1 was needed for phenotypes of HOIP CKO. The results (revised Fig. 7h and

Supplementary Fig. 9l) showed that the mRNA levels of osteogenesis related markers (such as *Alp*, *Colla1* and *Osx*) and protein levels of SUMRF1 substrates (such as SMAD1/5, MEEK2) were both enhanced in HOIP CKO BMSCs, when compared to untreated groups. ALP staining of CKO BMSCs treated with A17 also confirmed the enhanced osteogenesis of HOIP CKO BMSCs when compared with untreated groups (revised Fig. 7i). All of these data suggest that the osteogenesis defects of HOIP CKO mice are dependent on linear ubiquitination of SMURF1.

6) Throughout the manuscript the authors need to tone down any conclusions of linear ubiquitination substrates (except in the case of ILK) since the authors mainly show PPI data and no ubiquitination assays under stringent conditions.

Response: We thank the reviewer very much for the kind comments and the insightful concerns. According to the reviewers' suggestion, we have provided the better data of ILK and SMURF1 linear ubiquitination with a strict condition in revised Figs. 4c, 4m, 5d-f, 5h, 5l, 7g and Supplementary Figs. 3e, 3f, 7d and 8g. These results supported our conclusions on linearly ubiquitinated substrates.

REVIEWERS' COMMENTS

Reviewer #1 (Remarks to the Author):

The authors answered all the questions, and the revised manuscript significantly improved. I have no concerns.

Reviewer #2 (Remarks to the Author):

The authors addressed the concerns raised in the first review

The analysis/representation of the results in Figures is easier to understand now and the results with the Shigella infection in Figure 4 are stronger with the CFU assay as performed.

No further comments.

Reviewer #3 (Remarks to the Author):

The authors have sufficiently answered all questions and concerns. No further revisions are requested.